# An acquired phosphatidylinositol 4-phosphate transport initiates T-cell deterioration and leukemogenesis

Wenbin Zhong[1,2], Weize Lin[1], Yingjie Yang[1], Dan Chen[1], Xiuye Cao[1], Mengyang Xu[1,3], Guoping Pan[1], Huanzhao Chen[1], Jie Zheng[1], Xiaoqin Feng[4], Li hua Yang[5], Chaofeng Lai[1], Vesa M. Olkkonen [6,7], Jun Xu[3], Shuzhong Cui [2] ✉ & Daoguang Yan [1,2] ✉

Lipid remodeling is crucial for malignant cell transformation and tumorigenesis, but the precise molecular processes involved and direct evidences for these in vivo remain elusive. Here, we report that oxysterol-binding protein (OSBP)-related protein 4 L (ORP4L) is expressed in adult T-cell leukemia (ATL) cells but not normal T-cells. In ORP4L knock-in T-cells, ORP4L dimerizes with OSBP to control the shuttling of OSBP between the Golgi apparatus and the plasma membrane (PM) as an exchanger of phosphatidylinositol 4-phosphate [PI(4)P]/cholesterol. The PI(4)P arriving at the PM via this transport machinery replenishes phosphatidylinositol 4,5-bisphosphate [PI(4,5)P$_2$] and phosphatidylinositol (3,4,5) trisphosphate [PI(3,4,5)P$_3$] biosynthesis, thus contributing to PI3K/AKT hyperactivation and T-cell deterioration in vitro and in vivo. Disruption of ORP4L and OSBP dimerization disables PI(4)P transport and T-cell leukemogenesis. In summary, we identify a non-vesicular lipid transport machinery between Golgi and PM maintaining the oncogenic signaling competence initiating T-cell deterioration and leukemogenesis.

Phosphoinositides (PIPs) are phosphorylated forms of phosphatidylinositol (PI) generated by a variety of PI and PIP kinases. These phospholipids present in minute amounts serve crucial roles in regulating virtually every cellular process within eukaryotes[1,2]. Acute metabolic changes in plasma membrane (PM) phosphoinositides, such as those mediating signaling reactions, are rapidly compensated by homeostatic responses whose molecular basis is incompletely understood. PI(4,5)P$_2$ is a key informational phospholipid, localized to and defining the inner leaflet of the PM. How PM PI(4,5)P$_2$ is sourced and regulated is critically important to the understanding of a variety of fundamental cellular processes including signal transduction and tumorigenesis[3–5].

PI(4)P pool serves as the precursor of the bulk of PI(4,5)P$_2$[6,7]. Different cellular pools of PI(4)P contribute to the generation of PM PI(4,5)P$_2$ and to the regulation of downstream signaling events[7–9]. In view of these considerations, and of numerous findings that link PI(4)P metabolism to human disease[9,10], it is critically important to understand the mechanisms controlling the dynamic equilibrium of PI(4)P in different membranes.

The organelle-specific localizations of PIPs are due to the activity of compartment-specific PI and PIP kinases as well as of PI transfer proteins (PITPs) and proteins capable of inter-membrane PIP transfer[2,11]. Recently, significant progress has been made in identifying

[1]MOE Key Laboratory of Tumor Molecular Biology, Jinan University, Guangzhou 510632, China. [2]Affiliated Cancer Hospital and Institute of Guangzhou Medical University, Guangzhou 510095, China. [3]Research Center for Drug Discovery, School of Pharmaceutical Sciences, Sun Yat-Sen University, Guangzhou 510006, China. [4]Hematology and Oncology, Nanfang Hospital, Southern Medical University, Guangzhou 510515, China. [5]Pediatric Hematology Department, Zhujiang Hospital, Southern Medical University, Guangzhou 510282, China. [6]Minerva Foundation Institute for Medical Research, Biomedicum 2U, FI-00290 Helsinki, Finland. [7]Department of Anatomy, Faculty of Medicine, University of Helsinki, FI-00014 Helsinki, Finland. ✉e-mail: Cuishuzhong@gzhmu.edu.cn; tydg@jnu.edu.cn

the PIP transfer proteins and characterizing their biology. The oxysterol-binding protein (OSBP) and OSBP-related proteins (ORPs) have emerged as key mediators and regulators of non-vesicular lipid transport at organelle membranes contact sites (MCSs), structures providing an ideal location at which PITPs and PIP transfer proteins can achieve the efficient transfer of lipids between organelles via non-vesicular mechanisms[12–14]. Besides this lipid transfer mode, some PITPs also undergo physical translocation or redistribution in facilitating PI transport to the PM[15,16]. Although the remodeling PIs in the PM is crucial for cancer cells[3–5], the precise molecular processes involved and direct evidences for their roles in initiating tumorigenesis are largely missing.

Enhanced PI3K/AKT signaling activity is considered a hallmark of cancer, which is dependent on PM $PI(4,5)P_2$ and $PI(3,4,5)P_3$ contents[17,18]. However, exactly how the dynamic equilibrium of PM PIPs is regulated and how these changes couple to subsequent activation or inactivation of signaling in tumorigenesis remain poorly understood[15]. Our recent study reported that ORP4L is required for human T-lymphotropic virus type 1 (HTLV-1) oncogene Tax-induced T-cell leukemogenesis[19]. We used ORP4L knock-out mice and demonstrated ORP4L deletion blocks Tax-induced T-cell leukemia. For molecular insight, HTLV-1 caused epigenetic loss of miR-31 resulting in the release of ORP4L expression. ORP4L interacted with and activated PI3Kδ, leading to $PI(3,4,5)P_3$ generation and AKT activation. However, how the homeostasis of PIPs in the PM is maintained and the substrate, especially PI(4)P, is supplied to support sustained robust PI3K/AKT signaling is unknown. In this work, we report a lipid transport mode in which ORP4L orchestrates together with OSBP the transport of PI(4)P from Golgi to PM, thereby coupling PI(4)P transport and PI3K/AKT oncogenic signaling to govern T-cell malignant transformation.

## Results

### PI(4)P at the Golgi contributes to rapid PM $PI(4,5)P_2$ replenishment following agonist stimulation in ORP4L KI T-cells

We previously demonstrated that ORP4L is expressed in T-cell acute lymphoblastic leukemia (T-ALL) cells but not normal T-cells[20] and is a prerequisite for T-cell leukemogenesis induced by HTLV-1[19]. In the present study we confirmed that ORP4L is highly expressed in adult T-cell leukemia (ATL) cells (Supplementary Fig. 1a). To further verify the role of ORP4L in T-cell leukemogenesis in vivo, we generated a T-cell-specific ORP4L knock-in (KI) mouse line by CRISPR/Cas9 genome editing in the Rosa26 locus, in which the coding sequences of human *ORP4L* are flanked by loxP sites (Supplementary Fig. 1b). In this allele, removal of the stop cassette flanked by loxP sites leads to ORP4L expression. After crossing with transgenic mice in which Cre is driven by T-cell specific Lck promoter, the mice expressed ORP4L specifically in T-cells (Supplementary Fig. 1c).

Our previous study demonstrated ORP4L regulates lipid composition of the PM[21]. Non-targeted lipidomics was conducted to examine the lipid profiles in the PM of ORP4L KI T-cells. A total of 192 lipid molecular species were identified and quantified (Fig. 1a, Supplementary Data 1), revealing changes in the concentrations of several phosphoinositides. ORP4L interacts with PI3Kδ to promote $PI(3,4,5)P_3$ generation and AKT activation in HTLV-1 induced T-cell transformation[19], but it is unknown how the dynamic equilibrium of PIs is controlled to support sustained $PI(3,4,5)P_3$ generation in the PM. The evidence for functions of ORPs in the intracellular transport and metabolism of phosphoinositides[22,23] promotes us to focus on $PI(4,5)P_2$ and its synthetic precursor PI(4)P, two important phospholipids that fulfill many crucial functions in the PM[8]. No change of PM PI(4)P (Fig. 1b) but a reduction of $PI(4,5)P_2$ was observed in the ORP4L KI T-cells (Fig. 1c), indicating the presence of an uncovered mechanism that maintains the equilibrium of PI(4)P and $PI(4,5)P_2$ in the PM of ORP4L KI T-cells. In the PM, PI(4)P can be synthesized from PI by PI4KIIIα, and $PI(4,5)P_2$ is produced from PI(4)P by PIP5KB[6] (Fig. 1d).

Therefore, the PI(4)P supply is critical for the maintenance of PM $PI(4,5)P_2$[24]. To determine the precursor sources of the PM $PI(4,5)P_2$, we monitored the recovery of GFP-$PH_{PLC\delta1}$ domain[25] as a real-time indicator of PM $PI(4,5)P_2$ production. Upon anti-CD3 treatment triggering PM $PI(4,5)P_2$ consumption, $PI(4,5)P_2$ levels declined steadily but recovered rapidly in the ORP4L KI T-cells. Depletion of PIP5KB prevented the recovery of $PI(4,5)P_2$, while depletion of PI4KIIIα showed only minor effects on the $PI(4,5)P_2$ levels (Fig. 1e; Supplementary Movie 1). These experiments show that PM PI(4)P is the major source for $PI(4,5)P_2$ resynthesis, but the PM PI pool is not used as a major precursor for PI(4)P synthesis; Indeed, PI(4)P may be additionally supplied from other membranes. Besides the PM, Golgi complex is another major site of PI(4)P biosynthesis. We therefore employed GFP-$P4M_{SidM}$ domain[26] as a real-time indicator of PM and Golgi PI(4)P. PM PI(4)P declined steadily and then rapidly recovered upon anti-CD3 treatment of ORP4L KI T-cells; This effect was accompanied by a continuous decline of Golgi PI(4)P levels (Fig. 1f; Supplementary Movie 2). PI4KIIIα knockdown did not affect the PI(4)P recovery in the PM and decline in the Golgi, indicating that PM $PI(4,5)P_2$ levels are to a large extent supplied by a PI(4)P precursor pool in the Golgi of ORP4L KI T-cells. To further confirm this, we conducted PI4KIIα knockdown experiments to reduce PI(4)P generation in the Golgi of ORP4L KI T-cells. We found depletion of PI4KIIα prevented the recovery of $PI(4,5)P_2$ upon anti-CD3 stimulation (Fig. 1g), suggesting PI(4)P generated in the Golgi is the main source of $PI(4,5)P_2$ replenishment of PM in ORP4L KI T-cells.

### OSBP translocates from the Golgi to the PM via dimerizing with ORP4L

We next studied the mechanism of PI(4)P transport from Golgi to PM in the ORP4L KI T-cells. Transmission electron microscopy revealed that in ORP4L KI T-cells and normal T-cells, the gap between the *trans* face of Golgi and the PM was much narrower than that in the adherent HepG2 cells and 293 T cells (Fig. 2a). A physical interaction of ORP4L with OSBP was observed in ORP4L KI T-cells (Fig. 2b). We found that OSBP localizes at the Golgi complex of ORP4L KI T-cells, where OSBP and Golgi are in close proximity of the PM, displaying a degree of overlap with PM marker staining (Fig. 2c).

OSBP markedly alters its localization upon binding of its oxysterol ligand[27], but ORP4L displays stable localization at the PM of T-cell leukemia cells[20,28]. In ORP4L KI T-cells, anti-CD3 stimulation significantly enhanced the interaction of ORP4L and OSBP (Fig. 2b); this stimulation induced translocation of OSBP to the PM where it co-localized with ORP4L (Fig. 2d). We isolated PM and Golgi membrane fractions and verified their purification (Supplementary Fig. 1d). Cell fractionation analysis provided further evidence for translocation of OSBP from the Golgi to the PM upon anti-CD3 stimulation of the ORP4L KI T-cells, but not in wild-type T-cells (Fig. 2e). To directly visualize the interaction of ORP4L and OSBP in vivo, we employed the bimolecular fluorescence complementation (BiFC) assay. Expression vectors encoding ORP4L/pVn-C1 and OSBP/pVc-C1 or ORP4L/pVn-C1 and OSBP/pVc-N1 were co-transfected into ORP4L KI T-cells (Fig. 2f). Some heterodimer fluorescence was observed in the PM of cells transfected with ORP4L/pVn-C1 and OSBP/pVc-C1, and this fluorescence signal was enhanced when the cells were stimulated with anti-CD3 (Fig. 2g), supporting a specific interaction of ORP4L and OSBP in vivo.

Furthermore, the translocation of OSBP to the PM was blocked in ORP4L KI T-cells upon ORP4L knockdown (Fig. 2h, i). We overexpressed the wild-type OSBP or a mutant lacking the ORP4L binding site (OSBP△ORP4L)[29] in ORP4L KI T-cells, revealing that the wild-type OSBP translocated to the PM along with ORP4L, but OSBP△ORP4L failed to do this (Fig. 2j). In ORP4L KI T-cells, VAPA knockdown did not affect the redistribution of OSBP in the PM (Fig. 2k). We constructed OSBP constructs without a FFAT motif (OSBP△FFAT) to further study

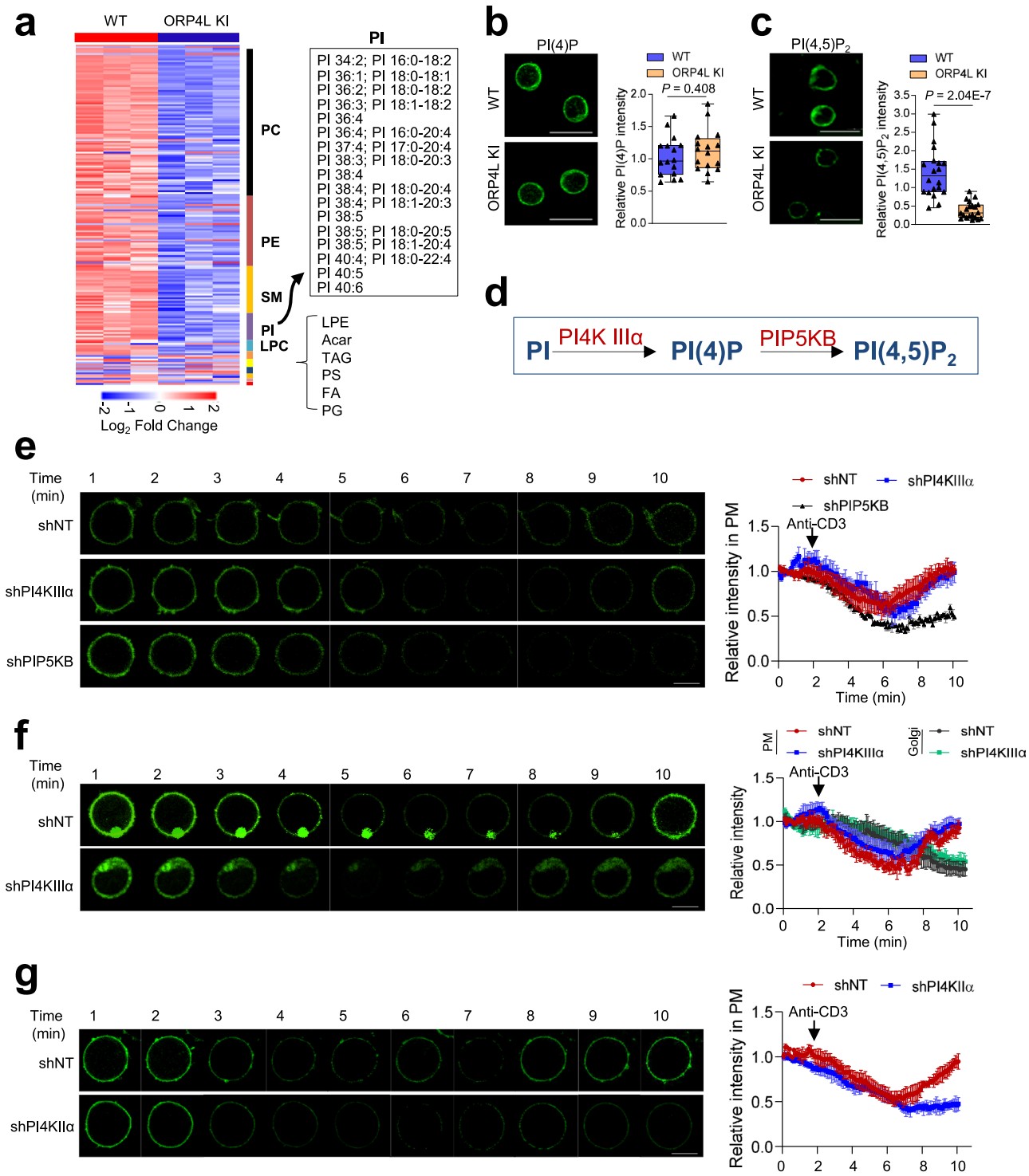

**Fig. 1 | PM phosphoinositides are remodeled and PI(4)P in the Golgi is transported to the PM in ORP4L KI T-cells. a** Heatmap of PM lipid changes in T-cells of ORP4L KI and littermate wild-type mice (*n* = 3 mice). **b, c** PI(4)P (**b**) and PI(4,5)P$_2$ (**c**) contents in T-cells of ORP4L KI and littermate wild-type mice. Scale bar, 10 µm. The right panel indicates quantitation of relative fluorescence intensity. Black triangles present in dividual data points for *n* = 16 (panel b) and *n* = 20 (panel c) cells from one mouse. The same experiment was repeated twice in T-cells from two mice. **d** Schematic representation of PIPs metabolism in the PM. **e** Time course of PI(4,5)P$_2$ probe GFP-PH$_{PLC\delta1}$ at the PM of ORPL4 KI T-cells subjected to PI4KIIIα or PIP5KB knockdown. The right panel indicates quantification of relative fluorescence intensity changes of GFP-PH$_{PLC\delta1}$ at the PM. Scale bar, 5 µm. **f** Time course of the PI(4)P probe GFP-P4M$_{SidM}$ at PM and Golgi complex of ORPL4 KI T-cells subjected

to PI4KIIIα knockdown. The right panel indicates quantification of relative fluorescence intensity changes of GFP-P4M$_{SidM}$ at the PM (upper) and Golgi (lower). Scale bar, 5 µm. **g** Time course of the PI(4,5)P$_2$ probe GFP-PH$_{PLC\delta1}$ at the PM of ORPL4 KI T-cells subjected to PI4KIIα knockdown. The right panel indicates quantification of relative fluorescence intensity changes of GFP-PH$_{PLC\delta1}$ at the PM. Scale bar, 5 µm. In panel (**e–g**), the data are presented as Mean ± SD (*n* = 10 cells from one mouse). The same experiments were repeated twice in T-cells from two mice. In each box plot of (**b, c**), the central mark indicates the median, the bottom and top edges of the box indicate the interquartile range, and the whiskers represent the maximum and minimum point. Two-tailed unpaired *t*-test with a confidence interval of 95% was used to compute statistics. *P*-values are indicated in the figures. Source data are provided as a Source Data file.

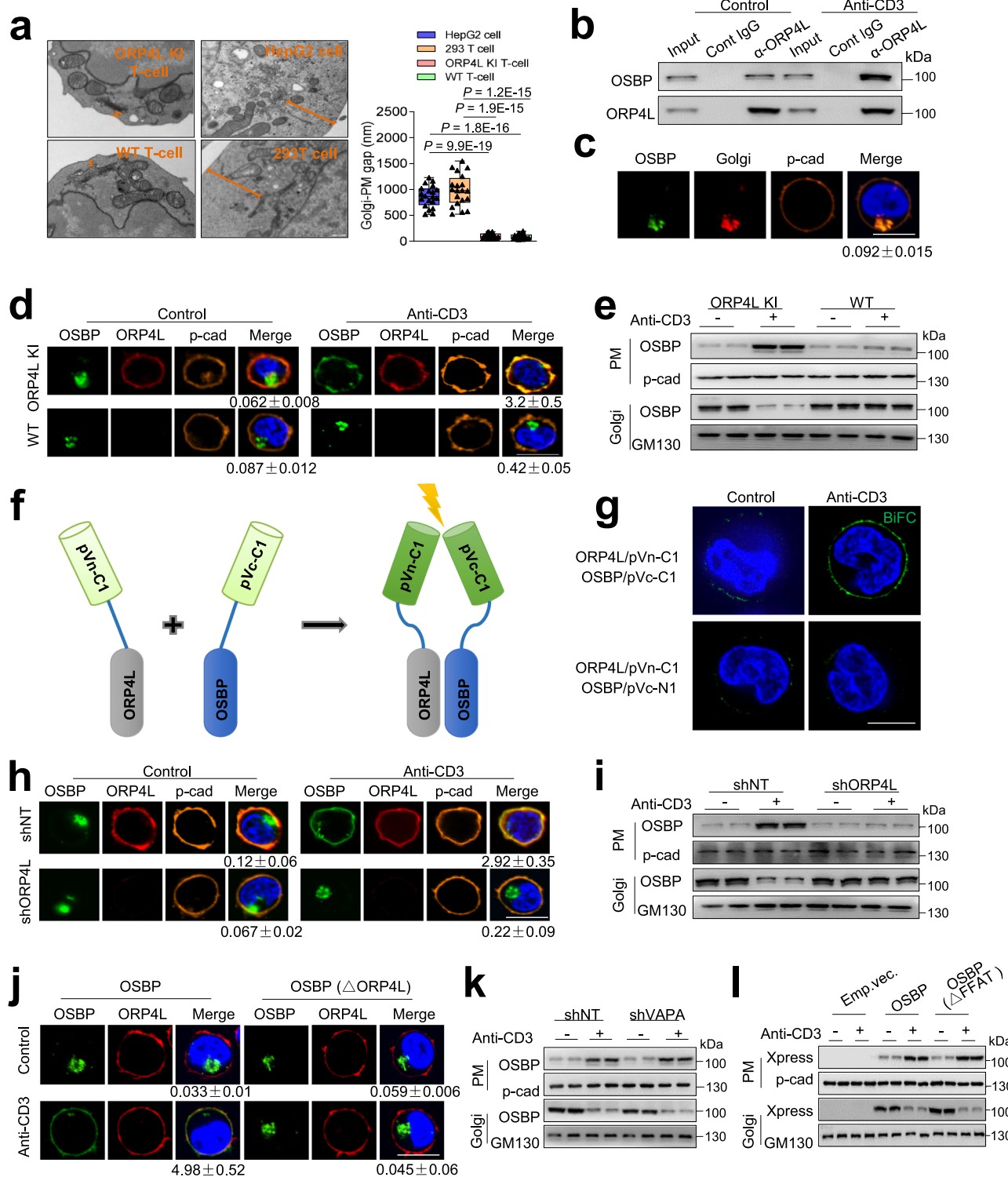

the VAPA dependency of the OSBP translocation. Similar to wild-type OSBP, also OSBP△FFAT was translocated from the Golgi to the PM upon anti-CD3 stimulation (Fig. 2l). These results provide evidence that OSBP translocates from the Golgi to the PM in ORP4L KI T-cells, depending on the interaction with ORP4L, independent of VAPA.

### ORP4L dimerizes with OSBP to establish a machinery for PI(4)P transport from Golgi to PM

The above observation promoted us to hypothesize that PI(4)P may act as cargo transported by OSBP between the Golgi and the PM. Given that both OSBP and ORP4L bind to PI(4)P, we proposed that these two

proteins may display different binding affinity. We purified the ORP4L and OSBP proteins (Supplementary Fig. 1e) for surface plasmon resonance (SPR) assays, revealing that OSBP preferentially binds PI(4)P as compared to ORP4L (Fig. 3a). We therefore measured the kinetics of PI(4)P and OSBP at the Golgi and PM compartments in ORP4L KI T-cells. PM PI(4)P levels declined and recovered after anti-CD3 stimulation, the process being associated with a steady decrease of Golgi PI(4)P (Fig. 3b, c; Supplementary Movie 3). Coinciding with recovery of the PM PI(4)P to pre-stimulation levels, OSBP translocated from Golgi to the PM (Fig. 3b, c; Supplementary Movie 4), indicating that the translocation of OSBP and transport of PI(4)P from Golgi to PM occur

**Fig. 2 | OSBP translocates from Golgi to PM via interacting with ORP4L.**
**a** Electron micrographs of ORP4L KI and wild-type T-cells, HepG2 and 293 T cells. Scale bar, 100 nm. Black triangles present in dividual data points for *n* = 20 cells from one mouse or cell line. The central mark indicates the median, the bottom and top edges of the box indicate the interquartile range, and the whiskers represent the maximum and minimum point. One-way ANOVA test with a confidence interval of 95% was used to compute statistics. **b** Co-immunoprecipitation analysis of ORP4L binding to OSBP in ORP4L KI T-cells with or without anti-CD3 stimulation. **c** The localization of OSBP in ORP4L KI T-cells. Scale bar, 10 μm. **d** The localization of OSBP in ORP4L KI and wild-type T-cells with or without anti-CD3 stimulation. Scale bar, 10 μm. **e** Western blot analysis of OSBP protein levels in PM and Golgi of ORP4L KI or wild-type T-cells. **f** A schematic representation describing the BiFC technique. **g** Interactions between ORP4L/pVn-C1 and OSBP/pVc-C1 in ORP4L KI

T-cells as determined by BiFC. Scar bar, 10 μm. **h** The localization of OSBP in ORP4L KI T-cells upon ORP4L knockdown. Scar bar, 10 μm. **i** Western blot analysis of OSBP protein levels in PM and Golgi of ORP4L KI T-cells upon ORP4L knockdown. **j** The localization of overexpressed wild-type OSBP or OSBP (△ORP4L) in ORP4L KI T-cells. Scar bar, 10 μm. **k** Western blot analysis of OSBP protein levels in PM and Golgi of ORP4L KI T-cells with or without VAPA knockdown. **l** Western blot analysis of overexpressed wild-type OSBP or OSBP(△FFAT) protein levels in the PM and Golgi of ORP4L KI T-cells. Confocal microscopy and blot images are representative of *n* = 3 biological replicates with similar results. The same experiments were repeated twice in T-cells from two mice. In (**c**, **d**, **h** and **j**), the ration of OSBP fluorescence density from *n* = 10 cells from one mouse between PM and Golgi are shown below. Source data are provided as a Source Data file.

in a synchronized manner. Extended time monitoring revealed that after translocation to the PM, OSBP shuttled back to the Golgi, coincident with cholesterol monitored with an mCherry-D4H probe[30], which decreased in the PM but increased in the Golgi (Fig. 3b, c; Supplementary Movie 5). These findings indicate that OSBP circulates between Golgi and PM, mediating PI(4)P/cholesterol exchange. Consistently, OSBP knockdown reduced the PI(4)P/cholesterol exchange between PM and Golgi and the recovery of PM PI(4,5)P$_2$ after anti-CD3 stimulation (Fig. 3d). Experiments in which OSBP was overexpressed in the knock-down cells (OSBP expression was confirmed in Supplementary Fig. 1f) revealed that wild-type OSBP rescued the post-stimulation reduction of PM PI(4)P and recovery of PI(4,5)P$_2$. However, OSBP with the PI(4)P or ORP4L binding sites mutated[12,29] failed to do this (Fig. 3d). We knocked down ORP4L and then re-expressed ORP4L with OSBP binding site mutations (ORP4L△OSBP) in the ORP4L KI T-cells (ORP4L expression was confirmed in Supplementary Fig. 1g). Strikingly, knocking down ORP4L inhibited the PM PI(4)P/cholesterol transport and PI(4,5)P$_2$ recovery. Wild-type ORP4L rescued these processes, but ORP4L△OSBP failed to do so (Fig. 3e). The above findings suggest that OSBP transports PI(4)P from Golgi to PM in exchange for cholesterol, dependent on its interaction with ORP4L.

PI(4)P transport over ER-Golgi contact sites by OSBP requires the anchor protein VAPA[12]. Similar to the OSBP translocation, in ORP4L KI T-cells, VAPA knockdown did not affect the recovery of PI(4)P (Fig. 3f) in the PM. Consistently, OSBP△FFAT rescued the PI(4)P shuttling in OSBP knockdown cells (Fig. 3g, Supplementary Fig. 1h). In ORP4L knockdown cells, ORP4L△FFAT rescued the PI(4)P shuttling similar to wild-type ORP4L (Fig. 3h, Supplementary Fig. 1i), indicating that the above processes mediated by OSBP and ORP4L are likely independent of VAPA-mediated membrane contacts in this specific cell type.

We next employed OSW-1, an inhibitor of OSBP[31]. Pre-treatment with OSW-1 prevented OSBP translocation to the PM (Supplementary Fig. 1j); We verified that OSW-1 binds with OSBP, at a $K_D$ of 1.8 ± 0.37 μM (Supplementary Fig. 1k). OSW-1 competed with PI(4)P for binding to OSBP as analyzed by SPR ABA infection assay (Supplementary Fig. 1l, m), and inhibited the PI(4)P/cholesterol exchange (Fig. 3i, j). Taken together, our results show that ORP4L controls the redistribution of OSBP from Golgi to the PM, cargoing PI(4)P to the PM for conversion to PI(4,5)P$_2$.

Given that PI4KIIIβ fuels OSBP by spatial proximity[32], we tested whether ORP4L acts in PI(4)P transfer not only by interacting with OSBP but maybe also through regulating PI4KIIIβ or PI4KIIα spatial proximity. Co-immunoprecipitation experiments failed to reveal interaction of PI4KIIIβ or PI4KIIα with ORP4L in the presence or absence of anti-CD3 stimulation in ORP4L KI T-cells (Supplementary Fig. 1n). In addition, both PI4KIIIβ and PI4KIIα localized in the Golgi of ORP4L KI T-cells, and did not translocate to the PM upon anti-CD3 stimulation (Supplementary Fig. 1o). Thus, we concluded that ORP4L does not regulate PI4KIIIβ or PI4KIIα spatial proximity in ORP4L KI T-cells. Because ORP4L KI affected bulk lipid species in the PM of T-cells (Fig. 1a), we further studied whether sphingolipids are involved

in ORP4L-PI(4)P replenishment of PI(4,5)P$_2$ upon anti-CD3 stimulation in ORP4L KI T-cells. We treated ORP4L KI T-cells with serine, a precursor for all sphingolipids[33,34], and D609, a sphingomyelin synthase inhibitor[35,36] to increase and decrease sphingomyelin (SM) levels, respectively. We found the serine and D609 treatments did not change the PI(4)P transport and PI(4,5)P$_2$ replenishment upon anti-CD3 stimulation (Fig. 3k, l). To further clarify whether the reduction of PI(4)P and PI(4,5)P$_2$ replenishment upon ORP4L knockdown could result from a change in SM, we treated ORP4L knockdown cells with serine or D609. These two treatments could not rescue the PI(4)P transport and PI(4,5)P$_2$ recovery after anti-CD3 stimulation (Fig. 3m, n). These results indicated that SM is not involved in regulating the ORP4L/OSBP mediated PI(4)P transport at Golgi/PM interface of ORP4L KI T-cells.

We speculated that a close spacing between Golgi and PM in specific cell types could facilitate the PI(4)P transport function of OSBP. We therefore compared the role of OSBP in different cell types. In adherent cells, HepG2 and 293 T cells, ORP4L is mainly cytosolic without PM localization (Supplementary Fig. 2a). To further verify whether OSBP also can shuttle in adherent cells, we monitored PM PI(4,5)P$_2$ and OSBP translocation in HepG2 and 293 T cells. Upon treatment with histamine or EGF (both cell lines have receptors for these)[37–39] to trigger PM PI(4,5)P$_2$ consumption, PI(4,5)P$_2$ levels declined and recovered in these two cells type. Whereas, the OSBP localized in Golgi, without translocation or shuttle to PM throughout the ligands stimulation (Supplementary Fig. 2b). Meanwhile, we isolated Golgi and PM fractions of the cells after the ligands treatments. Western blot showing almost undetectable OSBP in PM, accompanied with stable OSBP levels in Golgi before or after ligand stimulation (Supplementary Fig. 2c–e). In ORP4L KI T-cells, OSBP overexpression increased the PM PI(4)P and PI(4,5)P$_2$ contents (Supplementary Fig. 2f), while its knockdown reduced them in the PM (Supplementary Fig. 2g). By contrast, in 293 T and HepG2 cells, although OSBP overexpression promoted PI(4,5)P$_2$ enrichment in the PM, it reduced the levels of PM PI(4)P (Supplementary Fig. 2f), while OSBP knockdown increased the PM PI(4)P (Supplementary Fig. 2g). The OSBP phenotype in these adherent kidney and liver cell lines is thus opposite to the ORP4L KI T-cells, suggesting that the pathways responsible for PIP transport and metabolism show marked heterogeneity between cell types.

## ORP4L knock-in mice develop T-cell leukemia
We next monitored the state of the hematopoietic system in the ORP4L KI animals: No abnormality in blood smears (Supplementary Fig. 3a), total white blood cell (WBC) count (Supplementary Fig. 3b), CD3$^+$ T-cell, CD19$^+$ B-cell and Gr-1$^+$ granulocyte count (Supplementary Fig. 3c) at the age of 6 or 12 months was observed. However, ORP4L KI mice developed a lethal phenotype beginning at the age of 15 months, and they all died within 30 months (Fig. 4a). These morbid ORP4L KI mice had a significantly higher total WBC count than wild-type animals (Fig. 4b). Flow cytometry revealed that CD3$^+$ T-cells were significantly increased in ORP4L KI animals, with normal CD19$^+$ B-cell and Gr-1$^+$ granulocyte cell count (Supplementary Fig. 3d). We characterized the

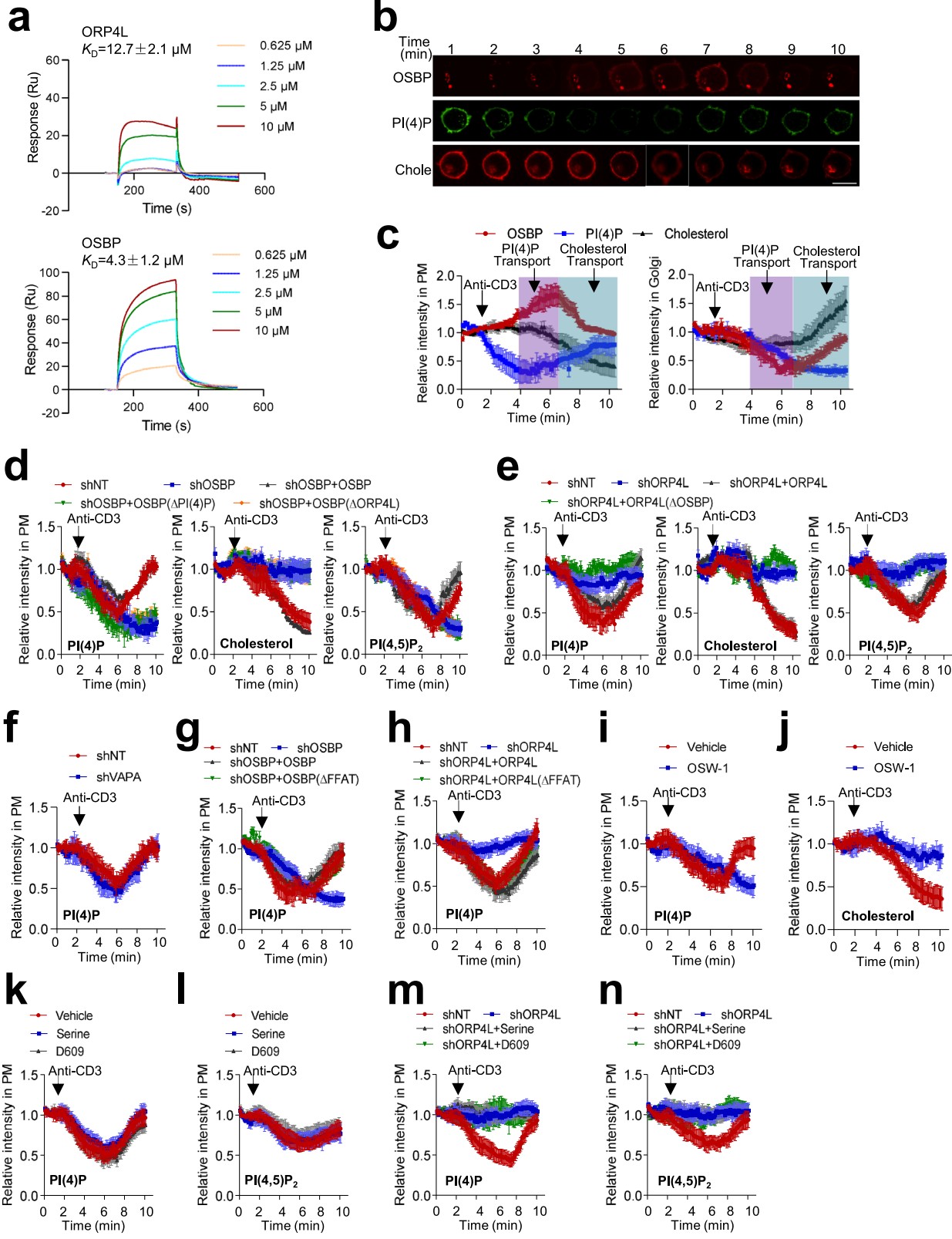

major phenotype of the ORP4L KI T-cells, revealing that they were CD4[+] T-cells. Further analysis showed that these cells included CD44[+]CD25[+] and c-Kit[+] T-cells (Fig. 4c). We further characterized the leukemia cells by analyzing their Foxp3 and CCR4 expression. The percentage of CD25[+]Foxp3[+] and CD25[+]CCR4[+] T-cells were increased in ORP4L KI mice (Fig. 4d). The Foxp3 and CCR4 expression were also upregulated in ORP4L KI T-cells (Fig. 4e).

Peripheral blood smears from KI mice showed the presence of leukemic cells with cleaved nuclei morphology, morphologically identical to the 'flower cells' in human ATL (Fig. 4f). ORP4L KI mice presented with the development of marked splenomegaly (Fig. 4g) and hepatomegaly (Fig. 4h). Histological analysis revealed CD3[+] T-cells infiltration in the spleen, liver, kidney, and lung (Fig. 4i, Supplementary Fig. 3e), and the examination of bone marrow revealed an abundant

**Fig. 3 | ORP4L/OSBP transfer PI(4)P from Golgi to PM for PI(4,5)P$_2$ biosynthesis.**
**a** Kinetic analysis of PI(4)P binding to ORP4L (upper) and OSBP (lower) determined by SPR in real-time. Mean $K_D$ values ± SD ($n = 3$ biological replicates) are indicated. **b**, **c** Time course of DsRed-OSBP protein, the PI(4)P probe GFP-P4M$_{SidM}$ and the cholesterol probe mCherry-D4H at the PM and Golgi of ORP4L KI T-cells. Quantification of fluorescent signals in the PM (**c**, left) and Golgi (**c**, right) are shown. Scale bar, 5 μm. **d** Relative fluorescence intensity changes of the GFP-P4M$_{SidM}$, mCherry-D4H and GFP-PH$_{PLCδ1}$ in the PM of ORP4L KI T-cells subjected to OSBP knockdown and re-expression with wild-type OSBP or OSBP(△PI(4)P) or OSBP(△ORP4L). **e** Relative fluorescence intensity changes of GFP-P4M$_{SidM}$, mCherry-D4H and GFP-PH$_{PLCδ1}$ in the PM of ORP4L KI T-cells subjected to ORP4L knockdown and re-expression of wild-type ORP4L or ORP4L(△OSBP). **f** Relative fluorescence intensity changes of GFP-P4M$_{SidM}$ in the PM of ORP4L KI T-cells upon VAPA knockdown.

**g**, **h** Relative fluorescence intensity changes of GFP-P4M$_{SidM}$ in the PM of ORP4L KI T-cells subjected to OSBP knockdown and re-expression with wild-type OSBP or OSBP(△VAPA) (**g**), or subjected to ORP4L knockdown and re-expression with wild-type ORP4L or ORP4L(△VAPA) (**h**). **i**, **j** Relative fluorescence intensity changes of GFP-P4M$_{SidM}$ (**i**) and mCherry-D4H (**j**) in PM of ORP4L KI T-cells pre-treated with or without 5 nM OSW-1 for 1 h. **k**, **l** Relative fluorescence intensity changes of GFP-P4M$_{SidM}$ (**k**) and GFP-PH$_{PLCδ1}$ (**l**) in PM of ORP4L KI T-cells pre-treated with or without 0.5 mM serine and 200 μM D609 for 12 h. **m**, **n** Relative fluorescence intensity changes of the GFP-P4M$_{SidM}$ (**m**) and GFP-PH$_{PLCδ1}$ (**n**) in PM of ORP4L KI T-cells subjected to ORP4L knockdown and then treated with or without 0.5 mM serine and 200 μM D609 for 12 h. In panel (**c–n**), data are presented as Mean ± SD ($n = 10$ cells from one mouse). The same experiments were repeated twice in T-cells from two mice. Source data are provided as a Source Data file.

presence of lymphomatous cells (Fig. 4i). In addition, the ORP4L KI mice displayed cutaneous pathology with gross skin disease, revealing marked hyperkeratosis and acanthosis, characterized by CD3$^+$ T-cells infiltrated into the epidermis (Fig. 4i, Supplementary Fig. 3e), which is also characteristic of human ATL[40]. To validate whether malignant clones of the lymphomatous cells of the KI mice are able to transfer the disease, we injected splenocytes from individual ORP4L KI animals into B-NDG mice; these mice displayed prominent splenomegaly and lymphomatous infiltration with T-cell leukemia phenotype (Fig. 4j, k), thus providing evidence for the leukemogenic potential of the ORP4L KI T-cells.

## ORP4L promotes PI(3,4,5)P$_3$ generation, AKT activation and gene expression reprogramming

As PI(4,5)P$_2$ is consumed as a second message to downstream signaling pathways, we explored the role of PM phosphoinositide remodeling by ORP4L on downstream signaling events that facilitate malignant transformation of T-cells. As reported in our previous study[19], the catalytic product of PI3Kδ, PI(3,4,5)P$_3$ was significantly elevated in ORP4L KI T-cells (Fig. 5a). AKT activation is linked to NF-κB activation, p53 inhibition and cell survival in ATL[41]. Consistent with the increased PI(3,4,5)P$_3$ production, ORP4L KI T-cells displayed elevated AKT and NF-κB activation as well as inhibition of p53 (Fig. 5b). In ORP4L KI T-cells and MT-4 cell line, AKT inhibition by LY294002 resulted in suppression of p65 and p53 phosphorylation; Meanwhile, NF-κB inhibition by TPCA-1 resulted in suppression of p53 phosphorylation (Supplementary Fig. 4a), suggesting that enforced ORP4L expression leads to phospholipid dysregulation resulting in AKT and NF-κB activation and p53 inhibition.

Leukemias typically have the lowest numbers of mutations among different cancers[42], and it is now generally accepted that abnormal gene expression may be the key to initiating the tumorigenesis[43,44]. We executed whole genome sequencing of the ORP4L KI T-cells, but no significant oncogenic mutations characterized in ATL[45] were found (Supplementary Fig. 5a, b). To determine whether specific gene sets are dysregulated in the ORP4L KI T-cells, we performed whole transcriptome RNA sequencing. A total of 970 differentially expressed genes (DEGs) were detected, of which 539 were downregulated and 431 upregulated (Fig. 5c). KEGG pathway analysis of the differentially expressed genes highlighted the "transcriptional misregulation in cancer" pathway as the most extensively affected (Fig. 5d, e); Several genes in this pathway, such as Cebpe[46], Erg[47], Igf1[48], Mmp9[49], Mpo[50], Pax5[51], Prom1[52] are involved in leukemia. These data indicate an ORP4L-dependent gene reprogramming upon the T-cells deterioration.

To confirm the connection of PI(4)P transport and the AKT-NFκB-p53 pathway, we silenced OSBP in the ORP4L KI T-cells. OSBP knockdown reduced the PI(3,4,5)P$_3$ content of the PM, coinciding with significant dampening of the AKT-NF-κB activation and p53 inhibition, the effects being rescued by overexpression of wild-type OSBP, but not the mutants lacking the ORP4L or PI(4)P binding sites (Fig. 5f, g).

Consistently, ORP4L knockdown also reduced the PI(3,4,5)P$_3$ content and AKT-NFκB-p53 pathway activity, the effects being rescued by overexpression of wild-type ORP4L, but not the ORP4L△OSBP mutant (Fig. 5h, i). It was also of high interest to understand the consequences of enforced ORP4L expression in pre-malignant T-cells. Increase of PI(3,4,5)P$_3$ (Supplementary Fig. 4b), as well as accelerated AKT-NF-κB-p53 pathway activation (Supplementary Fig. 4c) were detected in pre-malignant T-cells of 4-month-old ORP4L KI mice, an age at which transformed T-cells were not yet detectable, indicating the PI(4)P transport and PI3K/AKT pathway activation upon ORP4L expression in T-cells occur early during leukemia development, before emergence of the actual pathology. These results suggested that PI(4)P transport from Golgi to the PM by OSBP is an essential prerequisite for efficient PI(3,4,5)P$_3$ generation and AKT activation in ORP4L KI transformed T-cells.

Our previous study revealed that ORP4L extracted PI(4,5)P$_2$ for PLCβ3 catalysis[20,21]. To clarify whether the observed AKT activation is dependent on PLCβ3, we conducted genetic modification of ORP4L and PLCβ3 in ORP4L KI T-cells and MT-4 cell line. We found ORP4L knockdown reduced AKT phosphorylation (Supplementary Fig. 4d), but PLCβ3 knockdown did not changed this status (Supplementary Fig. 4e). Moreover, in rescue experiments, both wild-type ORP4L and mutant ORP4L without PLCβ3 binding site[53] could abolish the reduction of AKT phosphorylation upon ORP4L knockdown in MT-4 cells (Supplementary Fig. 4f). Thus, we concluded the AKT activity is not dependent on PLCβ3 in this leukemia cell type.

## PI(4)P transport from Golgi to the PM contributes to T-cell deterioration in vivo

To verify that PI(4)P transport from Golgi to the PM contributes to T-cell transformation, we studied the role of ORP4L△OSBP, disabled to transport PI(4)P, in T-cell deterioration in vitro and in vivo. We transfected normal T-cells with lentivirus encoding wild-type ORP4L or ORP4L△OSBP (Fig. 6a) and cultured them in vitro. After culture for 16 weeks, T-cells transduced with wild-type ORP4L highly expressed Foxp3 and CCR4, as compared to control or ORP4L△OSBP transduced T-cells (Fig. 6b). HTLV-1 infected T-cells exhibit an initial phase of IL-2-dependent growth; over time, the cells become IL-2 independent[54]. After culture for 16 weeks, T-cells with wild-type ORP4L expression continued to proliferate following the removal of IL-2 from the media, but T-cells expressing ORP4L△OSBP died rapidly (Fig. 6c). The T-cells with wild-type ORP4L, but not the ORP4L△OSBP, displayed reduced levels of PI(4,5)P$_2$ (Supplementary Fig. 4g), increased PI(3,4,5)P$_3$ (Supplementary Fig. 4h) and activation of the AKT-NF-κB-p53 pathway (Supplementary Fig. 4i). Transplantation of wild-type ORP4L expressing, but not the ORP4L△OSBP expressing T-cells into B-NDG mice resulted in T-cell leukemia and death of the mice within 8 weeks (Fig. 6d). The mice transplanted with wild-type ORP4L expressing T-cells displayed a similar phenotype as the ORP4L KI animals (Fig. 6e–g). We next evaluated whether ORP4L deletion can

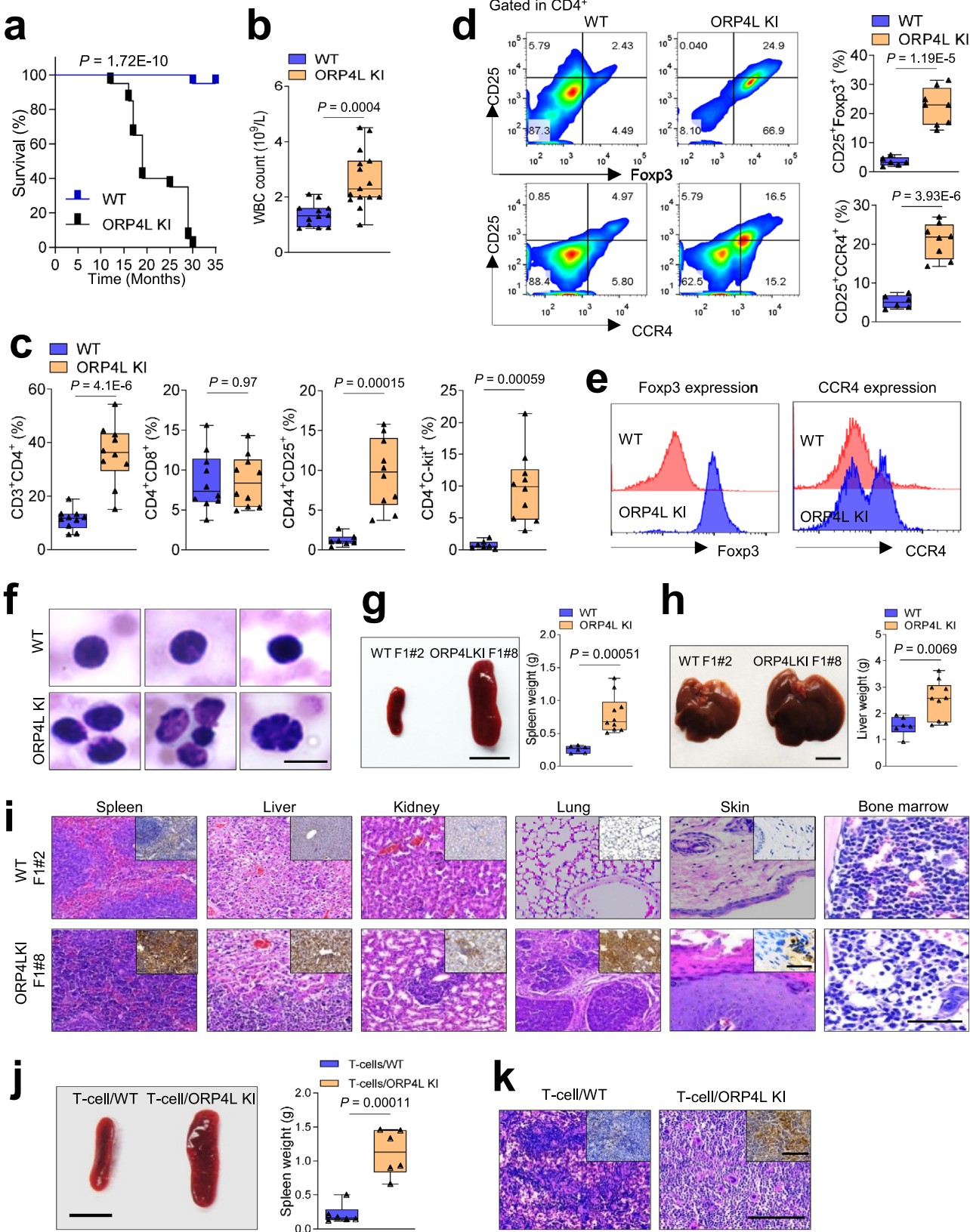

abolish leukemia formation induced by ORP4L KI T-cells. We constructed a doxycycline (dox)-inducible CRISPR/Cas9 system to knockout ORP4L as we recently described[19]. Induction of ORP4L sgRNA in ORP4L KI T-cells with Dox led to efficient reduction of ORP4L protein and AKT-NF-κB-p53 pathway activity (Supplementary Fig. 4j). We then transplanted these ORP4L sgRNA-carrying ORP4L KI

T-leukemia cells into B-NDG mice; ORP4L deletion by Dox induction significantly reduced T-leukemia cell engraftment (Fig. 6h) and leukemia formation (Fig. 6i), as well as improved the survival of the mice (Fig. 6j). These results demonstrated that PI(4)P transport from Golgi to the PM by the ORP4L/OSBP dimer drives T-cell deterioration and leukemia formation.

**Fig. 4 | Pathological findings of T-cell leukemia in ORP4L KI mice.**
**a** Kaplan–Meier comparative survival analysis of ORP4L KI and WT littermate mice ($n = 20$ mice per group, Log-rank test). **b** White blood cell count on peripheral blood of ORP4L KI and WT littermate mice. Black triangles present in dividual data points for $n = 15$ (ORP4L KI) and $n = 12$ (WT) mice. **c** The percentage of indicated cells in peripheral blood of ORP4L KI and WT littermate mice. Black triangles present in dividual data points for $n = 10$ mice. **d** Percentage of CD25$^+$Foxp3$^+$ or CD25$^+$CCR4$^+$ cells in peripheral blood of ORP4L KI and WT littermate mice. Black triangles present in dividual data points for $n = 8$ (ORP4L KI) and $n = 6$ (WT) mice. **e** Flow cytometry analysis of Foxp3 and CCR4 expression in ORP4L KI and WT littermate mice. The same experiments were repeated twice in two mice. **f** Representative blood smears of ORP4L KI and WT littermate mice ($n = 3$ mice). Scale bars, 10 μm. **g**, **h** Splenomegaly and hepatomegaly in ORP4L KI mice. The right panels illustrate the organ weights. Black triangles present in dividual data points for $n = 10$ (ORP4L KI) and $n = 6$ (WT) mice. Scale bars, 1 cm. **i** Representative H&E-staining of organs in ORP4L KI mice. Immunohistochemical anti-CD3 antibody staining (inserts) of the tissues is shown. Scale bars, 100 μm. **j** Gross and histological findings of splenomegaly in B-NDG mice after transplantation of spleen cells from ORP4L KI mice. Black triangles present in dividual data points for $n = 6$ mice. Scale bars, 1 cm. **k** Representative H&E and anti-CD3 antibody staining of spleen of **j**. Scale bars, 100 μm. Images of **i** and **k** are representative of $n = 3$ biological replicates with similar results. The same experiments were repeated twice in two mice. In each box plot, the central mark indicates the median, the bottom and top edges of the box indicate the interquartile range, and the whiskers represent the maximum and minimum point. Two-tailed unpaired $t$-test with a confidence interval of 95% was used to compute statistics. $P$-values are indicated in the figures. Source data are provided as a Source Data file.

## Discussion

It is poorly understood how lipid transport between organelle membranes affects cell fate decisions. In the present study, we provide evidence that ORP4L orchestrates OSBP to establish a PI(4)P transport route from Golgi to PM, contributing to hyperactivation of PI3K/AKT oncogenic signals and driving T-cells deterioration in vitro and in vivo (Fig. 6k). Our results identify an abnormal lipid transport route in malignant T-cells, which regulates the PM phosphoinositide pools, offering fresh insight into the mechanisms of cellular phosphoinositide dynamics and downstream oncogenic signaling events. More specifically, the present observations extend our understanding of the mechanisms underlying the initiation of T-cell deterioration and leukemia formation.

The PM is not a uniform structure, but it is composed of distinct domains that differ in lipid composition, structure, and signaling activity, and can be dynamically regulated by specific lipid-lipid and lipid-protein interactions[55,56]. One of the most significant properties of the neoplastic cells is altered lipid metabolism, and consequently, an abnormal cell membrane composition. In the transformed cells, bulk lipid species of the PM are reprogrammed to alter the physical properties of the PM and create a pro-tumor cellular state. For example, the changes of phosphatidylcholine result in the modulation of certain enzymes and accumulation of energetic material, which could be used for a higher proliferation rate[57]. Changes in phosphatidylserines could even be considered as cancer biomarkers[58,59]. We compared the PM lipidomics in normal T-cells and malignant transformed leukemia T-cells driven by enforced ORP4L expression. In these transformed T-cells, besides the PI, also the other lipids were reprogrammed to maintain the lipid homeostasis. In our previous study, we also found multiple lipid changes in the PM of leukemia stem cells upon ORP4L knockdown[21]. Consistent with this, lysophosphatidylcholine acyl-transferase LPCAT1 deletion significantly altered the bulk lipid composition and regulated oncogenic growth factor signaling of cancer cells[60].

Multiple lipid biosynthetic and trafficking pathways may contribute to the unique PM lipid composition of cancer cells[60]. Yet, the specific PITPs and PIP transporters responsible for mediating these PI/PIP reprogramming processes and how they promote tumorigenesis remain poorly understood. We demonstrate that PI(4)P at the Golgi rather than at the PM is important for the rapid replenishment of PI(4,5)P$_2$ following agonist stimulation of ORP4L KI T-cells. These results are consistent with earlier work suggesting that constitutive depletion of PI(4)P at the Golgi results in defective PI(4,5)P$_2$ replenishment[7]. We found that the ORP4L/OSBP heterodimer constitutes an intricate lipid transfer machinery that transports PI(4)P directly from the Golgi to the PM, by a process that is tightly coupled to PI(4,5)P$_2$ consumption and cholesterol transport in the opposite direction. The PIP change and subsequent AKT activation were detected in T-cells of young mice before the emergence of T-cell leukemia pathology, suggesting that this lipid transport machinery serves as an early, putatively initiating factor driving T-cell leukemogenesis in ORP4L KI T-cells.

OSBP and certain other ORP proteins have established roles in lipid exchange at ER-Golgi or ER-PM MCS[12–14]. Currently characterized eukaryotic lipid transport proteins are shuttles that typically extract a single lipid from the membrane of the donor organelle, solubilize it during transport through the cytosol, and deposit it in the acceptor organelle membrane or transporters that reach over membrane contact sites[61]. Blood cells are different from other adherent cell types, characterized by enlarged nuclei and compressed cytosolic space, resulting in a closer apposition between organelles. This cell type-specific feature also results in the different intracellular localization of ORP4L protein. In adherent cells, ORP4L is localized in ER and Golgi[62,63], whereas most of the ORP4L is found at the PM in leukemia cells[20,21] and macrophages[53]. Our data suggest that, in T-cells with a narrow Golgi-PM space, OSBP-mediated PI(4)P transport to the PM does not depend on MCS, but on a lipid transport mode in which OSBP protein shuttles with some oscillation over the narrow Golgi-PM gap, mediating cargo PI(4)P/cholesterol exchange. However, this OSBP shuttle and PI(4)P transport mode is absent in adherent cells. Alterations in cellular lipid metabolism act as a driving force modifying lipid transfer at organelle contact sites[23]; We suggest a model in which OSBP, ORP4L and PI3Kδ act in a sequential process that drives PI(4)P transport and metabolism, the oscillational shuttling of OSBP between Golgi and PM providing a sustained source of PI(4)P to promote PI(3,4,5)P$_3$ generation and AKT hyperactivation in ATL cells.

At this moment, we can only provide for our conclusion experimental evidence relying on in vivo data. We also wished to exploit an in vitro reconstituted membrane lipid transfer system employing lipid vesicles and purified proteins. However, we were thus far unable to reconstitute in vitro this non-vesicular lipid transport machinery. Although the gap between Golgi and PM is narrow, it is not close enough to form an MCS, and we find it possible that, in addition to ORP4L and OSBP, some other, as yet unknown proteins are required for the formation of the narrow Golgi-PM gap in this special cell type and the lipid transfer over it. Tethering proteins such as VAPA are needed in artificial liposomes reconstituting MCS[12], but our data suggest that VAPA is not involved in the present transfer mode. Thus, if tethering components are required for the PI(4)P-cholesterol transfer under study, they remain to be identified. The nature of the discovered process and the details of the transport mechanism are still being evaluated, but the present findings offer fresh insights into the concept of non-vesicular trafficking of lipids that drives T-cell deterioration.

## Methods

The Human Ethics Review Committee of Jinan University provided institutional approval of biosafety and ethical regulations for this study.

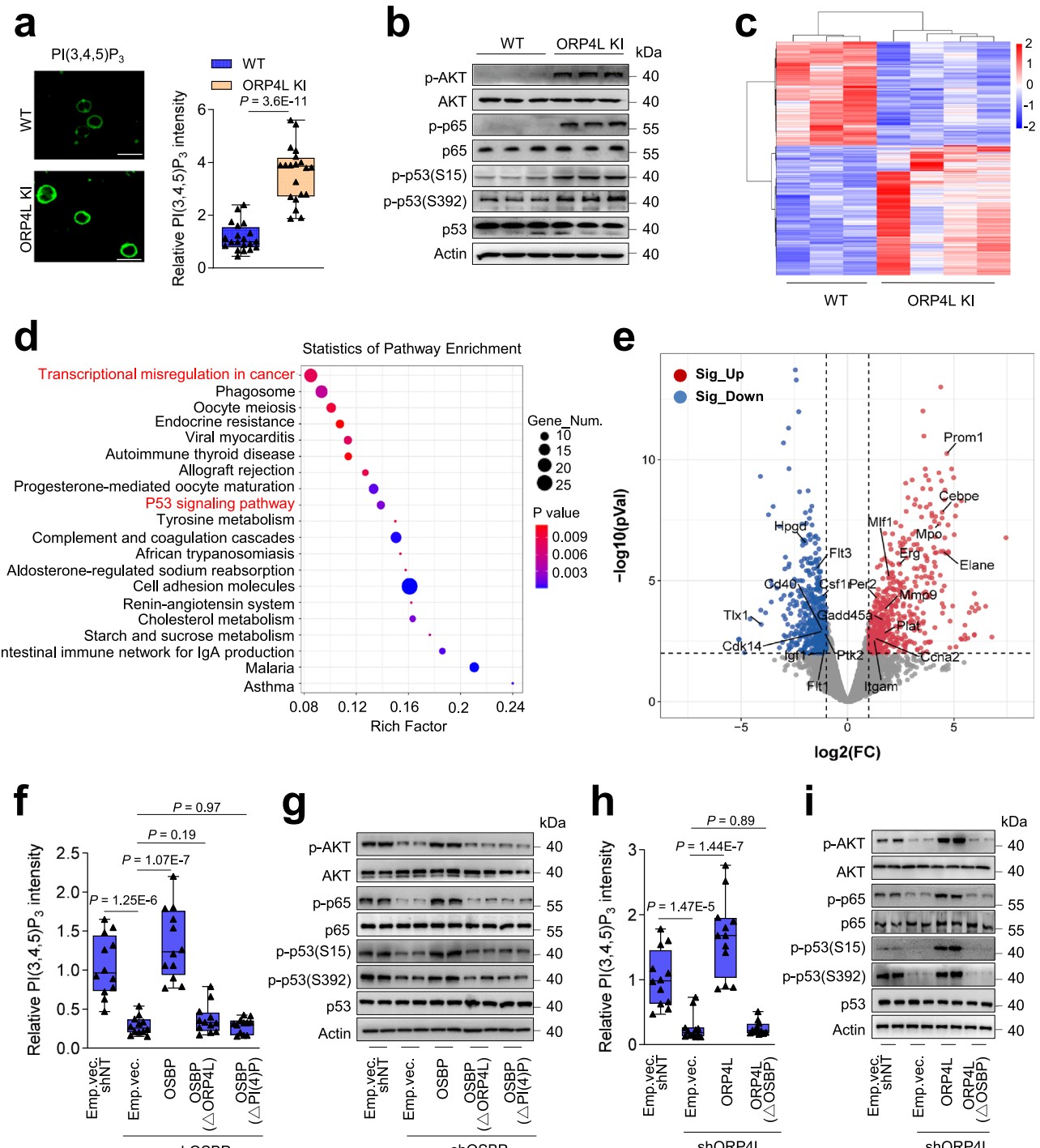

**Fig. 5 | ORP4L enhances PI(3,4,5)P₃ generation and AKT activation. a** PI(3,4,5)P₃ contents in T-cells of ORP4L KI and littermate wild-type mice. Scale bar, 10 μm. The right panels indicate quantitation of relative fluorescence intensity. Black triangles present in dividual data points for $n = 20$ cells from one mouse. **b** Phosphorylated AKT, p65 and p53 levels in T-cells of ORP4L KI and littermate wild-type mice. **c** Heatmap presentation of RNA sequencing data from ORP4L KI T-cells ($n = 4$ mice) at moribund state and wild-type littermate controls ($n = 3$ mice) at the same age. **d** KEGG signaling pathway enrichment of abnormally expressed genes for T-cells from ORP4L KI mice. **e** Volcano plot showing distribution of differentially expressed genes between ORP4L KI T-cells and wild-type littermate controls; The genes involved in "Transcriptional misregulation in cancer" are labeled. **f, g** The PI(3,4,5)P₃ content (**f**), phosphorylated AKT, p65 and p53 levels (**g**) in ORP4L KI T-cells subjected to OSBP knockdown and re-expression of wild-type OSBP or OSBP with PI(4)P and ORP4L binding site mutations. The panel (**f**) indicate quantitation of relative

fluorescence intensity. Black triangles present in dividual data points for $n = 12$ cells from one mouse. **h, i** The PI(3,4,5)P₃ content (**h**), phosphorylated AKT, p65 and p53 levels (**i**) in ORP4L KI T-cells subjected to ORP4L knockdown and re-expression of wild-type ORP4L or ORP4L with OSBP binding site mutations. The panel (**h**) indicate quantitation of relative fluorescence intensity. Black triangles present in dividual data points for $n = 12$ cells from one mouse. Blot images are representative of $n = 3$ biological replicates with similar results. In (**a**, **b**) and (**f–i**), the same experiments were repeated twice in T-cells from two mice. In each box plot, the central mark indicates the median, the bottom and top edges of the box indicate the interquartile range, and the whiskers represent the maximum and minimum point. Two-tailed unpaired $t$-test (**a**) and One-way ANOVA test (**f**, **h**) with a confidence interval of 95% was used to compute statistics. $P$-values are indicated in the figures. Source data are provided as a Source Data file.

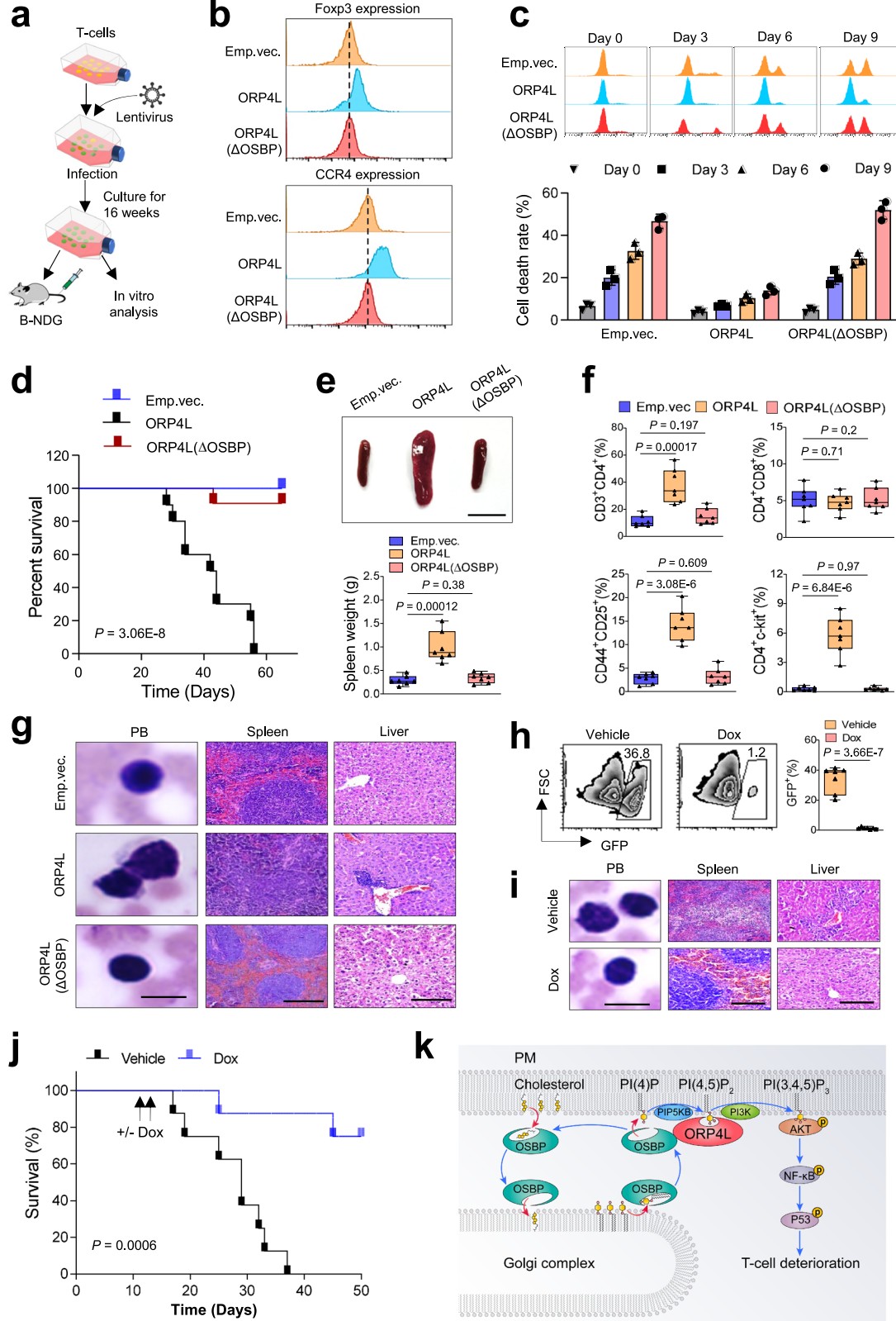

## Human specimens and cell lines

Six ATL patients and three healthy volunteers peripheral blood samples were collected after obtaining informed consent from Southern Medical University. The clinical information of subjects is provided in Supplementary Table 1. Peripheral blood mononuclear cells (PBMC) were purified by Ficoll-Hypaque gradient centrifugation. The T-cells were isolated by using Enhanced Human T Cell Immunocolumns

(Cedarlane) according to the manufacturer's instruction and cultured in RPMI 1640 containing 20% FBS and 20 U/mL recombinant human IL-2 (PeproTech), during which the medium was replaced every 3 days.

MT-4 cells (Cat# CL-0655), 293 T (Cat# CL-0005) and HepG2 cells (Cat# CL-0103) were purchased from Procell Life Science&Technology and maintained in RPMI 1640 (MT-4 cells) or DMEM (for 293 T and HepG2 cells) containing 10% FBS, 100 U/mL penicillin, and 100 mg/mL

**Fig. 6 | The role of PI(4)P transport from Golgi to the PM in T-cell deterioration.**
**a** Model outlining the experimental design. **b** Expression of Foxp3 and CCR4 in T-cells infected with lentivirus carrying ORP4L or ORP4L(ΔOSBP) and cultured for 16 weeks. **c** Cell death analysis of T-cells as treated in (**b**). Black shapes present individual data points for $n = 3$ biological replicates (Data are presented as Mean ± SD). **d** Kaplan−Meier comparative survival analysis of B-NDG mice transplanted with T-cells as treated in (**b**), ($n = 10$ mice per group, Log-rank test). **e** Splenomegaly in B-NDG mice treated as in (**d**). The lower panel illustrates the organ weights. Scale bars, 1 cm. **f** The percentage of indicated cells analyzed from B-NDG mice treated as in (**d**). **g** Representative peripheral blood smear and H&E-staining of spleen and liver from B-NDG mice treated as in (**d**). Scale bars, 10 μm for blood smear, 100 μm for H&E. **h** Percentage of GFP⁺ transplanted ORP4L KI T-cells in peripheral blood of B-NDG mice after 10 days treatment with or without Dox. **i** Representative peripheral blood smears and H&E-staining of spleen and liver in B-NDG mice treated as in (**h**). Scale bars, 10 μm for blood smear, 100 μm for H&E. **j** Comparative survival analysis of B-NDG mice treated with or without Dox ($n = 8$ mice per group, Log-rank test). **k** A model outlining the functions of ORP4L and OSBP heterodimer in PI(4)P transport in ORP4L KI T-cells. Images of (**g**, **i**) are representative of $n = 3$ biological replicates with similar results. The same experiments were repeated twice in two mice. In panel of (**e**, **f**, **h**), black triangles present individual data points for $n = 7$ mice. In each box plot, the central mark indicates the median, the bottom and top edges of the box indicate the interquartile range, and the whiskers represent the maximum and minimum point. One-way ANOVA test with a confidence interval of 95% was used to compute statistics. $P$-values are indicated in the figures. Source data are provided as a Source Data file.

---

streptomycin at 37 °C in a humidified incubator with 5% $CO_2$. The cell lines were authenticated by Promega Short-Tandem Repeat (STR) analysis and tested for mycoplasma contamination before experiments.

### Animals

All animal studies were conducted in accordance with regulations of the Institutional Animal Care and Use Committee of Jinan University (protocol no. 20180906-04). The animals were housed at Experimental Animal Center of Jinan University. Female B-NDG mice and both genders C57BL/6 J mice were used for experiments. All mice were housed in colony cages in a pathogen-free environment with the temperature maintained at 21−23 °C and relative humidity at 50−60%, and were under a 12 h light/12 h dark cycle.

### Generation of ORP4L knock-in (KI) mice

To generate a conditional ORP4L knock-in in Rosa26 locus, a targeting vector containing 5' homology arm, CAG promoter-loxp-stop-loxp-ORP4L-3x Flag-IRES-EGFP-WPRE-polyA fragment and 3' homology arms was obtained by using in-fusion cloning methods. The targeting vector was co-injected with Cas9 mRNA and gRNA into zygotes of C57BL/6 J to generate founder mice. Mice harboring the conditional allele were crossed with Lck-Cre transgenic mice (Jackson Stock No: 006889) in which Cre is driven by Lck promoter, to obtain mice that express ORP4L specifically in T-cells.

### cDNA constructs

Full-length human *ORP4L* (NM_030758.3) and *OSBP* (NM_002556) cDNAs were amplified by PCR and inserted into vector pcDNA™4His-MaxC (Invitrogen, Cat# V86420) for overexpression, or vector pGEX-4T-1 (Amersham, Cat# 27458001) for protein purification. The binding site mutant cDNAs were generated by PCR-based site-directed mutagenesis. Oligonucleotide primers used are listed in Supplementary Table 2.

### Gene transfer

High-titer lentiviral stocks (above $10^9$ TU/mL) carrying shRNA or cDNA were prepared by Shanghai GenePharma Co (Shanghai, China). For lentivirus infection, $1 × 10^6$ T-cells were resuspended in 100 μL medium containing lentivirus (multiplicity of infection, MOI = 100) and 5 μg/mL polybrene in 24-well culture plates. Infections were carried out for 6 h at 37 °C, 5% $CO_2$. After the end of infection, 400 μL medium was added. The T-cells were also electroporated using P3 Primary Cell 4D-Nucleofector™ X Kit L and 4D-Nucleofector™ X Unit (Lonza, Basel, Switzerland) according to the manufacturer's instructions. The shRNA sequences are provided in Supplementary Table 3. 293 T and HepG2 cells were transfected by using Lipofectamine 2000 (Thermo Fisher Scientific) according to the manufacturer's protocol.

### Mouse T-cell isolation and in vitro culture

Mouse T-cells were isolated from peripheral blood mononuclear cells (PBMC) or splenocytes. The whole peripheral blood was treated with RBC Lysis Buffer (BioLegend, Cat# 420301) to get PBMC. The spleens were digested with collagenase II (Sigma-Aldrich, Cat# 9001-12-1) at the concentration of 0.1 U/mL for 20 min at 37 °C, and the suspended cells were collected every 10 min, followed by centrifugation for 10 min at $300 × g$. After that, the red blood cells were removed by using RBC. CD4⁺ T-cells were further isolated from PBMC or splenocytes by using Naive CD4⁺ T Cell Isolation Kit (Miltenyi Biotec, Cat# 130-104-453) according to the manufacturer's instructions. After isolation, these T-cells were cultured in RPMI 1640 containing 20% FBS, 20 U/mL IL-2 (Peprotech, Cat# 212-12) and 1 μg/mL PHA (Sigma-Aldrich, Cat#693839).

### Lipidomics profiling by UHPLC-HRMS/MS

The plasma membrane was isolated by using Plasma Membrane Protein Isolation and Cell Fractionation Kit (Invent Biotechnologies, Inc.) according to the manufacturer's instructions. 300 μL of 50% aqueous methanol was added to the samples and they were transferred to 5 mL glass tubes and vortexed for 30 s. Twenty μL of the internal standards (5 μg/mL of FA 18:0-d₃₅, PC 15:0-18:1-d₇, PE 15:0-18:1-d₇, PG 15:0-18:1-d₇, TG 15:0-18:1-d₇−15:0 and 2 μg/mL of LPC 18:1-d₇, LPE 18:1-d₇, SM d18:1/18:1-d₉) were added. Then, 1 mL MTBE (methyl tert-butyl ether) was supplemented. The mixtures were vortexed for 1 min, ultrasonicated for 10 min, and stood for 10 min at 4 °C. After centrifugation at $3000 × g$ for 15 min, 800 μL supernatant (MTBE phase) was evaporated to dryness. The residues were reconstituted in 100 μL dichloromethane/methanol (1:1, v/v) prior to UHPLC-HRMS/MS analysis. Quality control (QC) sample was pooled from representative samples and analyzed with the same procedure as that for the experiment samples.

Chromatographic separation was performed on a Thermo Fisher Ultimate 3000 UHPLC system with a Waters CSH C18 column (2.1 mm × 100 mm, 1.7 μm) with column temperature at 55 °C. The flow rate was 0.3 mL/min. For positive mode: The injection volume was 1 μL. The mobile phases consisted of (A) acetonitrile/water (60:40, v/v) and (B) isopropanol/acetonitrile (90:10, v/v), both with 10 mM ammonium formate and 0.1% formate. A linear gradient elution was performed with the following program: 0 min, 40% B; 2 min, 45% B; 4 min, 55% B and held to 10 min; 14 min, 90% B; 15 min, 95% B and held to 18 min; 18.1 min, 40% B and held to 20 min. For negative mode: The injection volume was 2.5 μL. The mobile phases consisted of (A) acetonitrile/water (60:40, v/v) and (B) isopropanol/acetonitrile (90:10, v/v), both with 10 mM ammonium formate. A linear gradient elution was performed with the following program: 0 min, 40% B; 2 min, 45% B; 4 min, 55% B and held to 10 min; 14 min, 90% B; 15 min, 95% B and held to 18 min; 18.1 min, 40% B and held to 20 min.

The eluents were analyzed on a Thermo Fisher Q Exactive™ Hybrid Quadrupole-Orbitrap™ Mass Spectrome (QE) in Heated Electrospray Ionization Positive (HESI+) and Negative (HESI-) mode, respectively. Spray voltage was set to 3.5 kV for HESI+ and HESI−. Both Capillary and Aux Gas Temperature were 350 °C. Sheath gas flow rate was 40 (Arb). Aux gas flow rate was 10 (Arb). S-Lens RF Level was 50 (Arb). The full

scan was operated at a high-resolution of 70,000 FWHM (m/z = 200) at a range of 130–1950 m/z with AGC Target setting at $1 \times 10^6$. Simultaneously, the fragment ions information of top 10 precursors for each scan was acquired by Data-dependant acquisition (DDA) with HCD energy at 20, 30 and 40 eV, mass resolution of 17500 FWHM, and AGC Target of $5 \times 10^5$.

The raw data of UHPLC-HRMS/MS were firstly transformed to mzXML format by ProteoWizard and then processed by XCMS and CAMERA packages in R software platform. In XCMS package, the peak picking (method = centWave, ppm = 5, peakwidth = c(5, 20), snthresh = 10), alignment (bw = 6 and 3 for the first and second grouping, respectively), and retention time correction (method = obiwarp) were conducted. In CAMERA package, the annotations of isotope peak, adducts, and fragments were performed with default parameters. The final data was exported as a peak table file, including observations (sample name), variables (rt_mz), and peak areas. The area data of samples were corrected by the area data of QC samples. (Shen et al. 2016). The corrected data were normalized against total peak abundances before performing univariate and multivariate statistics.

For multivariate statistical analysis, the normalized data were imported to SIMCA software (version 14.1, AB Umetrics, Umeå, Sweden), where the data were preprocessed by Par scaling and mean centering before performing PCA, PLS-DA, and OPLS-DA. The model quality is described by the $R^2X$ or $R^2Y$ and $Q^2$ values. $R^2X$ (PCA) or $R^2Y$ (PLS-DA and OPLS-DA) is defined as the proportion of variance in the data explained by the models and indicates the goodness of fit. $Q^2$ is defined as the proportion of variance in the data predictable by the model and indicates the predictability of current model, calculated by cross-validation procedure. In order to avoid model over-fitting, a default 7-round cross-validation in SIMCA software was performed throughout to determine the optimal number of principal components.

For univariate statistical analysis, the normalized data were calculated by Student's *t*-test.

The variables with VIP values of OPLS-DA model larger than 1 and *p*-values of univariate statistical analysis <0.05 were identified as potential differential metabolites. Fold change was calculated as binary logarithm of average normalized peak intensity ratio between Group 1 and Group 2, where the positive value means that the average mass response of Group 1 is higher than Group 2.

Lipid species were quantified by dividing the area under the curve of each lipid by the area under the curve of its assigned internal standard. FA species—to the internal standard FA 18:0-d$_{35}$, PC species—to the internal standard PC 15:0-18:1-d$_7$, PE species—to the internal standard PE 15:0-18:1-d$_7$, PE species—to the internal standard PG 15:0-18:1-d$_7$, TG species—to the internal standard TG 15:0-18:1-d$_7$–15:0, LPC species—to the internal standard LPC 18:1-d$_7$, LPE species—to the internal standard LPE 18:1-d$_7$ and SM species—to the internal standard SM d18:1/18:1-d$_9$. The abundance of lipid species was then normalized to the cell numbers of each sample.

## Live cell imaging and PIPs quantification

Cells were transfected with the PI(4,5)P$_2$ probe GFP-PH$_{PLC\delta1}$, the PI(4)P probe GFP-P4M$_{SidM}$, the cholesterol probe mCherry-D4H or DsRed-OSBP. Imaging was carried out at 37 °C approximately 18 h after transfection. Before imaging, cells were transferred to imaging buffer (136 mM NaCl, 25 mM KCl, 2 mM CaCl$_2$, 1.3 mM MgCl$_2$, 10 mM HEPES, pH 7.4) that had been prewarmed to 37 °C. The images of cells were excited with low-intensity 488-nm or 546-nm laser excitation and acquired at 20-s intervals alternately under time-lapse mode with an Olympus FV3000 Confocal Microscope. The baseline fluorescence was collected 100 s before the ligand stimulation (10 μg/mL anti-CD3 for ATL cells, 10 μg/mL histamine, and 50 ng/mL EGF for HepG2 cells and 293 T cells). For comparisons of fluorescence intensities in the PM, the intensities at the cell circumference were measured using WCIF Image J software, and determined by averaging the intensity of 30 different point of each cell. For comparisons of fluorescence intensities in Golgi, the intensities were determined by normalizing the Golgi total fluorescence intensity with Golgi area. Image data were subsequently presented as a ratio of F/F0 in the final results, where F0 represents baseline fluorescence intensity in each cell. The experiment was repeated more than three times, and at least 10 cells were measured in each experiment.

## Electron microscopy

Cells were fixed and scanned by Transmission Electron Microscopy (TEM) according to routine TEM Staining Protocol. Briefly, cells were collected and fixed with 2.5% glutaraldehyde in 0.1 M PB (pH 7.4) for 4 h at room temperature. Pipette off fixative, tight cell pellet was washed in 0.1 M PBS 3 × 15 min and post-fixes with 1% OsO4 in 0.1 M PBS for 1.5 h. After pipetting off OsO4 and rinse in 0.1 M PBS 3 × 10 min. 50%, 70%, 95% ethanol, 95% (1:1, ethanol: acetone), 95% acetone, 100% acetone were used for cell pellets dehydration. Then cell pellets with acetone were replaced with 1:1 then 2:1 (Eponate 812: acetone) for 2 h, dried in desiccator with top off, and added into Eponate 812 at 37 °C overnight, baked in 60 °C oven for 48 h. Finally, cell pellets were ultrathin sectioned, collected onto grids, and stained with uranyl acetate, rinsed with distilled water and analyzed by Transmission Electron Microscopy (FEI TECNAI 10 TEM, PHILIPS). The space between the Golgi and the PM was counted from higher magnification images of these cells.

## Co-immunoprecipitations

We isolated T-cells from ORP4L KI mice. $2 \times 10^7$ cells were washed three times with 4 °C PBS and broken in lysis buffer (50 mM Tris-Cl, 150 mM NaCl, 0.5 mM MgCl$_2$, 10% glycerol, and 0.5% Triton X-100, pH 8.0) supplemented with Protease Inhibitor Cocktail (Roche Group, Cat# 11836170001). In some studies, cells were stimulated with 10 ug/mL anti-CD3 for 10 min at 4 °C before lysis. Lysates were centrifuged at $12,000 \times g$ for 30 min at 4 °C. With a portion of the total cell lysate collected to run as the input, the remaining lysate was incubated at 4 °C for 1 h with Protein G agarose (Thermo Fisher Scientific, Cat# 20398). The recovered supernatant was incubated overnight with 5 μg ORP4L (Sigma-Aldrich, Cat#HPA021514) antibody. For the negative control, supernatant was incubated with control IgG. Beads were centrifuged at $300 \times g$ for 5 min at 4 °C, washed three times with cold lysis buffer, and the samples in beads were eluted with SDS-PAGE loading buffer at 95 °C for 5 min, followed by analysis by immunoblotting.

## Immunofluorescence

For protein staining, cells seeded onto coverslips were fixed with 4% paraformaldehyde for 30 min at room temperature, followed by permeabilization with 0.1% Triton X-100 for 5 min, and blocked with 10% FBS for 30 min at room temperature. Cells were then incubated with primary antibodies in 5% FBS at 4 °C overnight. After washing three times (10 min each) with PBS, cells were incubated with secondary antibodies at 37 °C for 30 min. Antibodies were used as following: For Fig. 2c, Goat Anti-OSBP (Thermo Fisher Scientific, Cat# PA5-18218, 1:100), Mouse Anti-pan-cadherin (Santa Cruz, Cat#sc-59876, clone: CH-19, 1:50); Alexa Fluor 488-Donkey Anti-Goat IgG (H + L) secondary antibody (Thermo Fisher Scientific, Cat# A32814, 1:200), Alexa Fluor 647-Goat Anti-Mouse IgG (H + L) secondary antibody (Thermo Fisher Scientific, Cat# A32728, 1:200). For Fig. 2d and Fig. 2h, Goat Anti-OSBP (Thermo Fisher Scientific, Cat# PA5-18218, 1:100), Rabbit Anti-ORP4L (Sigma-Aldrich, Cat#HPA021514, 1:200), Mouse Anti-pan-cadherin (Santa Cruz, Cat#sc-59876, clone: CH-19, 1:50); Alexa Fluor 488-Donkey Anti-Goat IgG (H + L) secondary antibody (Thermo Fisher Scientific, Cat# A32814, 1:200), Alexa Fluor 546 Goat Anti-Rabbit IgG

(H + L) secondary antibody (Thermo Fisher Scientific, Cat# A-11035, 1:200), Alexa Fluor 647-Goat Anti-Mouse IgG (H + L) secondary antibody (Thermo Fisher Scientific, Cat# A32728, 1:200). For Fig. 2j, Mouse anti-Xpress antibody (Thermo Fisher Scientific, Cat# R910-25, 1:200), Rabbit Anti-ORP4L (Sigma-Aldrich, Cat#HPA021514, 1:200); Alexa Fluor 488-Goat Anti-Mouse IgG (H + L) secondary antibody (Thermo Fisher Scientific, Cat# A32723, 1:200), Alexa Fluor 546 Goat Anti-Rabbit IgG (H + L) secondary antibody (Thermo Fisher Scientific, Cat# A-11035, 1:200). For Supplementary Fig. 1o, Rabbit Anti-ORP4L (Sigma-Aldrich, Cat#HPA021514, 1:200), Mouse Anti-PI4KIIIβ (Santa Cruz, Cat# sc-166822, 1:50); Mouse Anti-PI4KIIα (Santa Cruz, Cat# sc-390026, 1:50); Alexa Fluor 546 Goat Anti-Rabbit IgG (H + L) secondary antibody (Thermo Fisher Scientific, Cat# A-11035, 1:200); Alexa Fluor 488-Donkey Anti-Mouse IgG (H + L) secondary antibody (Thermo Fisher Scientific, Cat# A32723, 1:200). For Supplementary Fig. 2a, Rabbit Anti-ORP4L (Sigma-Aldrich, Cat#HPA021514, 1:200), Mouse Anti-pan-cadherin (Santa Cruz, Cat#sc-59876, clone: CH-19, 1:50); Alexa Fluor 546 Goat Anti-Rabbit IgG (H + L) secondary antibody (Thermo Fisher Scientific, Cat# A-11035, 1:200); Alexa Fluor 488-Donkey Anti-Mouse IgG (H + L) secondary antibody (Thermo Fisher Scientific, Cat# A32723, 1:200). The specimens were analyzed using Olympus FV3000 Confocal Microscope by using FV31S-SW software.

For immunostaining of PIPs, cells grown on coverslips were fixed by the addition of 1 mL of 4% PFA in PBS for 15 min, then washed three times with PBS containing 50 mM NH$_4$Cl, followed by permeabilization for 5 min with 20 mM digitonin in PBS. After three washes with PBS, cells were blocked for 60 min in PBS containing 5% normal goat serum (NGS). Primary antibodies of PI(4)P (Enzo Life Sciences, Cat# Z-P004, 1:100), PI(4,5)P$_2$ (Enzo Life Sciences, Cat# ADI-915-062, clone: KT10, 1:100) and PI(3,4,5)P$_3$ (Enzo Life Sciences, Cat# Z-P345, 1:100) diluted in PBS containing 5% NGS were incubated at 4 °C overnight. After three washes (10 min each) with PBS, the specimens were incubated with Alexa Fluor 488-Goat Anti-Mouse IgG (H + L) secondary antibody (Thermo Fisher Scientific, Cat# A32723, 1:200), in PBS containing 5% NGS for 45 min. Cells were then washed three times (10 min each) with PBS and analyzed using Olympus FV3000 Confocal Microscope. For comparisons of fluorescence intensities between different samples, images were collected during a single session at identical excitation and detection settings. Fluorescence intensities at the cell circumference were measured using WCIF Image J software. The intensity of fluorescence was determined by normalizing the intensity of 30 different point of each cell. The experiment was repeated more than three times, and at least 20 cells were measured in each experiment.

## Bimolecular fluorescence complementation (BiFC) assay
BiFC constructs using fragments derived from Venus were generated, full-length human *ORP4L* and *OSBP* cDNAs were inserted into pVn-N1, pVc-N1, pVn-C1 and pVc-C1 vectors[64]. The ORP4L/pVn-C1 and OSBP/pVc-C1, or ORP4L/pVn-N1 and OSBP/pVc-N1 vectors were co-transfected into ORP4L KI T-cells and culture for 24 h. Cells were stimulated with or without 10 ug/mL anti-CD3 for 5 min, then fixed with 4% paraformaldehyde for 15 min at 4 °C. The BiFC fluorescence was detected by using Olympus FV3000 Confocal Microscope.

## Plasma membrane and Golgi preparation isolation
The plasma membranes and Golgi were isolated by using Minute™ Plasma Membrane Protein Isolation and Cell Fractionation Kit (Invent Biotechnologies, Inc.) and Minute™ Golgi Apparatus Enrichment Kit (Invent Biotechnologies, Inc.), according to the manufacturer's instructions.

## Protein purification
*ORP4L* and *OSBP* cDNA cloned in pGEX-4T-1 were transformed into *E. coli* RosettaTM (DE3) (Novagen) for expression of a GST-fusion protein. The bacterial were cultured at 37 °C to OD600 0.5-1.0, followed by induction with 0.1 mM IPTG for 16–18 h at 18 °C. The bacterial were collected, and crude bacterial lysates were prepared by sonication in lysis buffer (50 mM Tris-Cl, 150 mM NaCl, and 1% Triton X-100, 1 mM PMSF, pH 8.0) in the presence of the protease inhibitor Cocktail (Roche Group) mixture. Bacterial lysates were centrifuged for 20 min at 12,000 × *g*, and the supernatants were used for further purification with the GST-Bind beads (Novagen). The beads were suspended in the cell lysate and incubated for 1 h at 4 °C with gently shaking. After washing twice with Cleavage Buffer (50 mM Tris, pH 8.0, 150 mM NaCl, 2.5 mM CaCl$_2$), the beads with GST-fusion protein bound were cleaved by thrombin (Sigma, T6634-250UN) in Cleavage Buffer for 1 h at room temperature to eliminate the GST tag. The eluent containing the thrombin cleaved protein were concentrated on Amicon Ultra (cut-off 50 kDa). The purified recombinant protein was stored at −80 °C in present of 10% glycerol. The purified protein was verified by SDS-PAGE.

## Surface plasmon resonance assay
SPR experiments were carried out by using a Biacore 8 K SPR instrument (GE Healthcare). Proteins were immobilized onto CM5 chips by amine coupling. The SPR instrument pipeline was first washed with 200 mL PBST buffer (phosphate buffered saline, pH 7.4, containing 0.005% Tween-20) and the chip equilibrated with PBST overnight. Compounds and proteins were prepared in PBST. The concentration of DMSO in the experiments was <0.1%. Experiments were performed at 25 °C at 30 μL min$^{-1}$ with high-performance injection. In Fig. 3a and Supplementary Fig. 1k, compounds binding was plotted in a 5-point dose–response curve with analyses run in a 1:5 dilution series at concentrations as indicated, with PBST as the reference. The dissociation constant, K$_D$, was calculated based on the K$_{on}$ and K$_{off}$ value. SPR competition assays of Supplementary Fig. 1l and m were performed by using the Biacore 8K A-B-A injects function. In Supplementary Fig. 1l, OSBP was immobilized on Sensor Chip. In red trace, 5 μM OSW-1 was injected as solution A, then the mixture containing 10 μM water-soluble diC8-PI(4)P (Enzo Life Sciences, Cat# P-4008) and 5 μM OSW-1 was injected as solution B, finally, 5 μM OSW-1 was injected as solution A again. In black trace, 5 μM OSW-1 was injected as solution A, then 5 μM OSW-1 was injected as solution B, finally, 5 μM OSW-1 was injected as solution A again. In Supplementary Fig. 1m, OSBP was immobilized on Sensor Chip. In red trace, 10 μM water-soluble diC8-PI(4)P was injected as solution A, then the mixture containing 10 μM diC8-PI(4)P and 5 μM OSW-1 was injected as solution B, finally, 10 μM diC8-PI(4)P was injected as solution A again. In black trace, 10 μM diC8-PI(4)P was injected as solution A, then 10 μM diC8-PI(4)P was injected as solution B, finally, 10 μM diC8-PI(4)P was injected as solution A again. The results were analyzed using Biacore 8 K evaluation software (GE Healthcare).

## Flow cytometry analysis
100 μL of peripheral blood sample was lysed by RBC Lysis Buffer, cells were collected and suspended in Cell Staining Buffer (BioLegend, Cat# 420201). Flow cytometry was performed performed on FACS Aria III (Becton Dickinson) using standard methods. For detection of surface antigens, cells were washed and stained with saturating amounts of antibodies conjugated with fluorescein in the presence of FcR-specific blocking antibody for 20 min on ice. Antibodies used were as follows: Alexa Fluor-488 anti-mouse Gr-1 (Biolegend, Cat# 108417, clone: RB6-8C5, 1:100), PE anti-mouse CD19 (Biolegend, Cat# 152408, clone: 1D3/CD19, 1:100), Alexa Fluor-647 anti-mouse CD3ε (Biolegend, Cat# 100322, clone: 145-2C11, 1:100), PE anti-mouse CD8a (Biolegend, Cat# 100708, clone: 53-6.7, 1:100), Alexa Fluor-488 anti-mouse CD4 (Biolegend, Cat# 100423, clone: GK1.5, 1:100), Alexa Fluor-647 anti-mouse c-Kit (Biolegend, Cat# 105818, clone: 2B8, 1:100), APC anti-mouse CD25 (Biolegend, Cat# 102012, clone: PC61, 1:100), PE/Cy5 anti-mouse/human CD44 (Biolegend, Cat# 103010, clone: PC61, 1:100), Alexa Fluor-647 anti-mouse/human Foxp3 (Biolegend, Cat#320013, clone:

150D, 1:100), PE anti-mouse/human CCR4 (Biolegend, Cat# 131203, clone: 2G12, 1:100). The data were analyzed by FlowJo_V10 software. The gating strategy are shown in Supplementary Fig. 6.

### T-cell transplantation into B-NDG mice
Four weeks old female B-NDG mice were purchased from BIOCYTO-GEN and housed at Experimental Animal Center of Jinan University. For Fig. 4j, spleens of 3 individual ORP4L KI or control wild-type mice were digested with collagenase II at the concentration of 0.1 U/mL for 20 min at 37 °C, and the suspended cells were collected every 10 min, followed by centrifugation at $300 \times g$ for 10 min. After that, the red blood cells were removed by using RBC Lysis Buffer. Splenocytes suspended in PBS containing 2% FBS were injected intravenously into B-NDG mice ($1 \times 10^6$ cells/mouse).

For Fig. 6a, we harvested wild-type CD4[+] T-cells and infected T-cells with lentivirus encoding wild-type ORP4L or ORP4L with OSBP binding site mutations. The cells were cultured for 16 weeks in the presence of 20 U/mL IL-2, after which the cells were collected and injected intravenously into lethally irradiated B-NDG mice ($3 \times 10^6$ cells/mouse in 100 μL PBS containing 2% FBS).

For Fig. 6h–j, ORP4L KI T-cells were infected with inducible sgRNA system[19], and then the cells were transplanted into B-NDG mice ($1 \times 10^6$ cells/mouse in 100 μL PBS containing 2% FBS), and then monitored for T-cell leukemia development. In order to delete ORP4L gene, Dox was administered via food pellets (625 mg/kg) 10 days after transplantation. Pathological analyses were conducted after Dox treatment.

All the mice were euthanized when maximal tumor cell burden reached 60% in peripheral blood.

### Tissue histology and peripheral blood analysis
For hematoxylin and eosin staining, tissues were directly fixed and embedded in OCT embedding medium (Sakura, Cat# 16-004004) and sliced by frozen slicer (Leica CM1850). The sections were stained with hematoxylin and eosin (Beyotime, Cat# C0105M) by using a standard protocol. Then, the stained sections were washed in 70% ethanol and analyzed.

For immunohistochemistry staining, the mouse tissues were washed with PBS and fixed overnight with 4% paraformaldehyde in PBS at 4 °C. The samples were dehydrated, embedded in paraffin and sectioned into 6 μm-thick transverse sections, and mounted on Superfrost Plus Microscope Slides. The sections were deparaffinized in xylene and rehydrated in graded ethanol. Antigen retrieval was performed in 0.01 M boiled citrate buffer (pH 6.0) for 10 min, using microwave heating (750 watts), and endogenous peroxidase was blocked in methanol containing 3% $H_2O_2$ for 10 min, and then sections were rinsed in PBS. Sections were incubated overnight with anti-mouse CD3-specific antibody (Wuhan Boster Biological Technology, Cat# PB9093, 1:100) at 4 °C. Subsequently, biotin conjugated donkey anti-rabbit IgG(H + L) (Wuhan Boster Biological Technology, Cat# BA1002, 1:100) were incubated for 30 min at 37 °C, followed by 30 min of incubation with Strep avidin-Biotin-Complex (SABC). Finally, the sections were stained with 3,3'-diaminobenzidine (DAB) and counterstained with hematoxylin.

For Wright's-Giemsa staining of peripheral blood smears, a thin blood smear was prepared on a clean and dry microscopic glass slide, air dried, and fixed out in absolute methanol for 10 min, followed by staining the methanol fixed smear with diluted Wright's-Giemsa stain for 5 min and three washes with PBS. All the sections and blood smears were imaged and analyzed by EVOS™ FL Auto system by using EVOS FL Auto 2 Imaging System Software (Thermo Fisher Scientific).

### RNA-seq library preparation, sequencing and analysis
RNA was extracted following the Trizol reagent (Invitrogen, Cat# 15596026) manual. RNA was precipitated in 1:1 isopropanol (v/v) and 1 μL glycogen at −20 °C overnight. mRNA library was constructed using VAHTS mRNA-seq V3 Library Prep Kit (Cellagen Technology, Cat# NR611) following the manufacturer's instructions. Libraries were sequenced on an Illumina NovaSeq 6000 sequencer for 318 cycles. Reads that passed the Illumina quality filters were kept for the subsequent analyses. Adapters were trimmed from the reads, and reads shorter than 17 nt were discarded. The reads were mapped to the mouse mRNA reference database using FANSe3 algorithm on Chi-Cloud NGS Analysis Platform (Chi-Biotech Co. Ltd., Shenzhen, China).

### Whole genome sequencing
A total amount of 0.5 μg DNA per sample was used as input material for the DNA library preparations. Sequencing library was generated using Truseq Nano DNA HT Sample Prep Kit (Illumina, Cat#FC-121-4003) following manufacturer's recommendations and index codes were added to each sample. Briefly, genomic DNA sample was fragmented by sonication to a size of 350 bp. Then DNA fragments were end polished, A-tailed, and ligated with the full-length adapter for Illumina sequencing, followed by further PCR amplification. After PCR products were purified, libraries were analyzed for size distribution by Agilent 2100 Bioanalyzer and quantified by real-time PCR (3 nM).

The clustering of the index-coded samples was performed on a cBot Cluster Generation System using Hiseq X PE Cluster Kit V2.5 (Illumina) according to the manufacturer's instructions. After cluster generation, the DNA libraries were sequenced on Illumina Hiseq platform and 150 bp paired-end reads were generated.

Valid sequencing data was mapped to the reference human genome (UCSC hg19) by Burrows-Wheeler Aligner (BWA) software[65] to get the original mapping results stored in BAM format. If one or one paired read(s) were mapped to multiple positions, the strategy adopted by BWA was to choose the most likely placement. If two or more most likely placements presented, BWA picked one randomly. Then, SAMtools[66] and Picard (http://broadinstitute.github.io/picard/) were used to sort BAM files and do duplicate marking, local realignment, and base quality recalibration to generate final BAM file for computation of the sequence coverage and depth. Mapping step was very difficult due to mismatches, including true mutation and sequencing error, and duplicates resulted from PCR amplification. These duplicate reads were uninformative and shouldn't be considered as evidence for variants. We used Picard to mark these duplicates for follow up analysis.

Samtools mpileup and bcftools were used to do variant calling and identify SNP, InDels. Control-FREEC was utilized to do CNV detection, while Crest was specialized for SV discovery. The somatic SNV was detected by muTect[67], the somatic InDel by Strelka[68], and the somatic structural variants (SV) by CREST. Control-FREEC was used to detect somatic CNV.

### IL-2-independent T-cell transformation assay
IL-2-independent T-cell transformation was assayed as described previously[69,70]. Briefly, mouse T-cells infected with lentivirus were cultured for 16 weeks in medium containing 20 U/mL IL-2, during which the medium was replaced every 3 days. 16 weeks later, the IL-2 was removed from the medium for further culture, and the cell death rates were quantified by using LIVE/DEAD™ Fixable Near-IR Dead Cell Stain Kit (Thermo Fisher, Cat# L10119).

### Western blot analysis
Total protein samples were mixed with loading sample buffer, boiled for 10 min, and subjected to SDS-PAGE followed by transfer onto PVDF membranes (Millipore, Cat# IPVH00010). After blocking and incubations of the membranes with primary antibodies and HRP-secondary antibody conjugates, the blots were developed by enhanced chemiluminescence (Millipore, Cat#WBKLS0500) by using Tanon Imager 5200 software. The antibodies were used as following: anti-ORP4L (Sigma-Aldrich, Cat#HPA021514, 1:1000), anti-OSBP (Thermo Fisher

Scientific, Cat# PA5-18218, 1:1000), anti-Phospho-AKT (Thr308) (Cell Signaling Technology, Cat# 9275, 1:1000), anti-AKT (Cell Signaling Technology, Cat# 9272, 1:1000), anti-phospho-p53 (S15) (Cell Signaling Technology, Cat# 9284, 1:1000), anti-phospho-p53 (S392) (Cell Signaling Technology, Cat# 9281, 1:1000), anti-pan-cadherin (Santa Cruz, Cat# sc-59876, clone: CH-19, 1:200), anti-Xpress (Thermo Fisher Scientific, Cat# R910-25, 1:3000), anti-Actin (Proteintech Group, Cat# 60008-1-Ig, clone: 7D2C10, 1:3000), anti-PLCβ3 (Santa Cruz, Cat# sc-133231, 1:200), anti-PI4KIIIβ (Santa Cruz, Cat# sc-166822, 1:200), anti-PI4KIIα (Santa Cruz, Cat# sc-390026, 1:200), anti-p65 (Cell Signaling, Cat# 8242, 1:1000), anti-phopho-p65 (Ser536) (Cell Signaling Technology, Cat# 3033, 1:1000), anti-Na/K ATPase (Cell Signaling Technology, Cat# 8242, 1:1000), anti-TGN38 (Cell Signaling Technology, Cat# 95649, 1:1000), anti-calnexin (Cell Signaling Technology, Cat# 2679, 1:1000), anti-cytochrome C (Cell Signaling Technology, Cat# 11940, 1:1000), Goat Anti-rabbit IgG (H + L), HRP conjugate antibody (Proteintech Group, Cat# SA00001-2, 1:3000), Goat Anti-mouse IgG (H + L), HRP conjugate antibody (Proteintech Group, Cat# SA00001-1, 1:3000), Rabbit Anti-Goat IgG (H + L), HRP conjugate antibody (Thermo Fisher Scientific, Cat# A27014, 1:3000). All the uncropped scans of the blots are provided in the Source Data file.

### Statistical and quantification analyses

Statistical calculations were performed using GraphPad Prism 8. All data is presented as mean ± S.D., unless otherwise indicated. The number of independent experiments ($n$) used for statistical analysis are indicated in the legends. All in vitro experiments were repeated at least three times, and in vivo experiments at least once. All comparisons were made by unpaired two-tailed Student's $t$-test or One-way ANOVA test if more than two groups were analyzed. The $P$-values are indicated in the corresponding figures.

### Reporting summary

Further information on research design is available in the Nature Research Reporting Summary linked to this article.

## Data availability

A reporting summary for this article is available as Supplementary Information file. The whole genome sequencing raw datasets generated in this study have been deposited in the NCBI gen bank under accession ID: PRJNA681414, and RNA sequencing raw datasets have been deposited in GEO under accession ID: GSE199958. The mass spectrometry data of lipidomics is provided in Supplementary Data 1. The oligonucleotide sequence and details of clinical samples used are provided in Supplementary Table 1–3. Time course video of PIPs, cholesterol and OSBP are provided in Supplementary Movie 1–5. The remaining data are available within the Article, Supplementary Information or Source Data file. Source data are provided with this paper.

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

## Acknowledgements

This work was supported by grants from NSFC, China (grant 32071280, 8177043 to D.Y.), from Major Research Program of Guangdong Science & Technology (grant 2017A030308002 to D.Y.), and from NSFC for Young Scientists of China (grant 31900548 to W.Z.).

## Author contributions

D.Y. and S.C. funded the research. D.Y. conceived and designed the experiments. W.Z. and W.L. performed the in vitro experiments with the assistance of Y.Y., D.C., X.C. and C.L.. G.P. and J.Z. established and maintained mouse lines. M.X. performed SPR experiments with the assistance of J.X.. S.C., X.F. and L.Y. collected clinical specimens and analyzed the data. H.C. performed bioinformatics analysis. W.Z., V.M.O., S.C. and D.Y. analyzed the data and wrote the paper.

## Competing interests

The authors declare no competing interests.
