## [Peer Review File · Nature Communications]

Title: An acquired PI(4)P transport initiates T-cell deteriorationREVIEWER COMMENTS

Reviewer #1 (Remarks to the Author); expert in non-vesicular lipid transport:

The manuscript from Zhong and collaborators describes a solid set of experiments suggesting the existence of a PI(4)P/sterol-exchange mechanism between the Golgi apparatus and the plasma membrane as a main source for PI(4,5)P₂ during AKT oncogenic signaling. The authors used a combination of cell biology, biochemistry and genetic approaches to conclude that i) PI(4)P of Golgi serves as a source for PI(4,5)P₂ biogenesis at plasma membrane, ii) ORP4L, together with OSBP, mediates PI(4)P transfer from the Golgi to plasma membrane, iii) ORP4L is involved in T-cell leukemia through PI(3,4,5)P₃ and AKT pathway activation. I found the results very convincing and supportive of the conclusions drawn in this paper. My enthusiasm for this work is only tempered by couple of points I would like to ask:

Major points:

- 1- Lipidomic analysis of ORP4L KI shows lipid modification in not only PI but also PC, PE and SM for instance. What the functional relevance of PC, PE and SM in ORP4L-mediated processes? It is well known that SM plays important function in OSBP-mediates ER-Golgi PI(4)P/sterol-exchange mechanisms. However, the present study does not address the involvement of sphingolipids (SL) in ORP4L-PI(4)P replenishment of PI(4,5)P₂ upon anti-CD3. Is the ORP4L-PI(4)P mechanism completely independent from SL in this cell type at Golgi/PM interface or are SLs involved in regulating the process?
- 2- I found the demonstration of the Golgi as being a source of PI(4)P for PI(4,5)P₂ regeneration at plasma membrane convincing, the use of PIP5KB and PI4KIII α mutants is useful to draw conclusion for PM-localizing enzymes at the genetic level. However, genetic evidence is lacking at the Golgi level. This could be reached through PI4KIII β or PI4KII α that both produce PI(4)P at the Golgi. A mutant for PI4KIII β or PI4KII α should in principle, if the assumption that Golgi is the main source of PI(4,5)P₂ replenishment, prevents the regeneration of PI(4,5)P₂ at PM. This would add a genetic evidence for Golgi PI(4)P.
- 3- In the same way, given that it was shown before that PI4KIII β fuel OSBP by spatial proximity, we can expect that ORP4L acts in PI(4)P transfer not only by interacting with OSBP but maybe also through regulating PI4KIII β spatial proximity. Was PI4KIII β ever tested in co-IPs experiments as in Fig. 2?
- 4- Interaction between OSBP and ORP4L was tested in co-IP. However, what is the in vivo evidence for direct interaction between OSBP and ORP4L? If in vivo experiment is lacking I suggest to be more cautious in the conclusion on OSBP and ORP4L interaction.
- 5- I found very convincing that PI(4)P exchange against sterols occurs between Golgi and PM. However this conclusion only relies on the levels of PI(4)P at Golgi and PM in mutants. An in vitro lipid transfer assay would definitely strengthen this major point of the paper if this is technically and timely feasible in the frame of this study.

6- A previous study from the group shows that ORP4L might act to better present PI(4,5)P₂ for catalysis by PLC at the ER/Golgi interface. In the present work, activation of the AKT pathway is suggested to act downstream of ORP4L in T-cell leukemia. It is known that PLC controls AKT phosphorylation. So, at Golgi/PM interface in this cell type, an important question will be to determine whether the phosphorylation status of AKT is changed in ORP4L mutants, and would it depend then on PLC?

Minor points

1- Oscillation or waves of OSBP/PI(4)P has been suggested to act at ER/Golgi interface, I found this was a bit missing in the discussion of the present study for the Golgi/PM interface in T-cell.

2- Is it possible to add some kind of quantification for Fig. 2c, d, f, h?

Reviewer #2 (Remarks to the Author); expert in ATL:

Zhong et al. reported that expression of ORP4L is enhanced in adult T-cell leukemia (ATL) cells, and ORP4L associates with OSBP, resulting in translocation of OSBP from the Golgi apparatus and the plasma membrane (PM) to exchange PI(4)P and cholesterol. PI(4)P at the PM leads to PI3K/AKT hyperactivation, and malignant transformation of T cells. This study shows that abnormal lipid transport from Golgi to PM causes activation of PI3K/AKT, and NFκB, and eventually development of T-cell leukemia. Thus, this study reveals important roles of lipid metabolism and malignant transformation of T cells.

Major concerns

1. The same authors have recently reported that ORP4L interacts with PI3Kδ to promote PI(3,4,5)P₃ generation and AKT activation in Blood. Authors should clarify the differences of these studies and discuss about discrepancies.
2. This study showed that leukemic cells from ORP4L KI mice are CD4⁺CD25⁺, which resemble to ATL cells. However, CD4⁺CD25⁺ means that these cells are similar to activated CD4⁺ T cells or regulatory T cells. ATL cells are derived from dysregulated regulatory T cells. Authors should analyze expression of Foxp3, CCR4, and CD45RO of leukemic cells from ORP4L KI mice.
3. The quality of blood smear is too poor (Figures 4 and 5). It is impossible to judge these cells as leukemic cells.
4. Different from leukemia observed in ORP4L KI mice, the feature of skin lesion in ATL is the presence of leukemic cells in the epidermis. Authors should clarify whether leukemic cells infiltrate into the epidermis.
5. Authors found that “transcriptional misregulation in cancer pathway” was affected in leukemic cells in ORP4L KI mice. However, since “transcriptional misregulation in cancer pathway” is too vague, authors should clarify whether dysregulated pathways in ATL (reference 38) are affected in this mouse model.
6. In line 247-249, “Human ATL is associated with elevated PI3K/AKT activity, resulting in oncogenic

properties through NF- κ B dependent p53 inhibition (reference 39 and 40)". These findings are not confirmed by the following studies. Therefore, it is controversial among HTLV-1 researchers.

7. It is not generally accepted that "Transition from IL-2-dependent to independent growth is a key step in the transformation T-cells (line 268-269)". Most of primary ATL cells do not proliferate by IL-2. This transition is observed only in in vitro culture system of ATL cells.

Minor concerns

1. In line 96-99, "promoting" should be "promotes".

Point-by-point responses to the reviewers' criticisms:

Reviewer #1 (Remarks to the Author); expert in non-vesicular lipid transport:

The manuscript from Zhong and collaborators describes a solid set of experiments suggesting the existence of a PI(4)P/sterol-exchange mechanism between the Golgi apparatus and the plasma membrane as a main source for PI(4,5)P₂ during AKT oncogenic signaling. The authors used a combination of cell biology, biochemistry and genetic approaches to conclude that i) PI(4)P of Golgi serves as a source for PI(4,5)P₂ biogenesis at plasma membrane, ii) ORP4L, together with OSBP, mediates PI(4)P transfer from the Golgi to plasma membrane, iii) ORP4L is involved in T-cell leukemia through PI(3,4,5)P₃ and AKT pathway activation. I found the results very convincing and supportive of the conclusions drawn in this paper. My enthusiasm for this work is only tempered by couple of points I would like to ask:

Major points:

1- Lipidomic analysis of ORP4L KI shows lipid modification in not only PI but also PC, PE and SM for instance. What the functional relevance of PC, PE and SM in ORP4L-mediated processes? It is well known that SM plays important function in OSBP-mediates ER-Golgi PI(4)P/sterol-exchange mechanisms. However, the present study does not address the involvement of sphingolipids (SL) in ORP4L-PI(4)P replenishment of PI(4,5)P₂ upon anti-CD3. Is the ORP4L-PI(4)P mechanism completely independent from SL in this cell type at Golgi/PM interface or are SLs involved in regulating the process?

Response: It is indeed important to discuss why the ORP4L KI affected bulk lipid species in the plasma membrane of T-cells. One of the most significant properties of the neoplastic cells is altered lipid metabolism, and consequently, an abnormal cell membrane composition. In this way, lipid changes act as a critical node integrating signaling with lipid remodeling to alter the physical properties of the plasma membrane and create a pro-tumor cellular state. For example, changes of phosphatidylcholine result in the modulation of certain enzymes and accumulation of energetic material, which could be used for a higher proliferation rate (Podo, F. et. al., *Frontiers in Oncology*, 2016). Furthermore, changes such in phosphatidylserines could even be considered as cancer biomarkers (Perrotti F. et al., *Int J Mol Sci.*, 2016). Additionally, some changes of the biophysical properties of cell membranes lead to higher resistance to chemotherapy, and finally to disturbances in signaling pathways (Wojciech Szlasa et. al., *Journal of Bioenergetics and Biomembranes*, 2020). We compared the plasma membrane lipidomics in normal T-cells and malignant transformed leukemia T-cells driven by enforced ORP4L expression. In these transformed T-cells, besides the PIs, also the other lipids of the plasma membrane were reprogrammed to maintain a new lipid homeostasis. In our previous study, we also found multiple lipid changes in the plasma membrane of leukemia stem cells upon ORP4L knockdown (Wenbin Zhong et. al., *Cell Reports*, 2018). Consistent with this, a lysophosphatidylcholine acyltransferase LPCAT1 deletion significantly altered the bulk lipid composition and regulated oncogenic growth factor signaling of cancer cells (Junfeng Bi et. al., *Cell Metabolism*, 2019). We further discuss this topic in the discussion section of new version.

In addition, as your suggested, we also performed new experiments to study whether sphingolipids are involved in ORP4L-PI(4)P replenishment of PI(4,5)P₂ upon anti-CD3 in ORP4L KI T-

leukemia cells. Firstly, we treated ORP4L KI T-cells by using exogenous serine, which is a precursor for all sphingolipids and increase SM level (CR Gault et.al., *Adv. Exp. Med. Biol.*, 2010; L. Ashley Cowart et. al., *J. Biol. Chem.*, 2007), or D609, a sphingomyelin synthase inhibitor reducing SM level (C. Luberto et. al., *J. Biol. Chem.*, 1998; Li Z. et al., *Biochim Biophys Acta*, 2007) to increase and decrease SM levels, respectively. We found serine and D609 treatments did not change the PI(4)P transport and PI(4,5)P₂ replenishment upon anti-CD3 stimulation (Figure 3k, l of new version) in ORP4L KI T-cells. As we previously showed, ORP4L knockdown reduced PI(4)P and PI(4,5)P₂ replenishment. To further clarify whether this reduction resulted from a change in SM, we treated ORP4L knockdown cells with serine or D609. These two treatments could not rescue the PI(4)P transport and PI(4,5)P₂ recovery after anti-CD3 stimulation (Figure 3m, n of new version). These results indicated that SM is not involved in regulating ORP4L/OSBP mediated PI(4)P transport at the Golgi/PM interface of ORP4L KI T-cells.

2- I found the demonstration of the Golgi as being a source of PI(4)P for PI(4,5)P₂ regeneration at plasma membrane convincing, the use of PIP5KB and PI4KIII α mutants is useful to draw conclusion for PM-localizing enzymes at the genetic level. However, genetic evidence is lacking at the Golgi level. This could be reached through PI4KIII β or PI4KII α that both produce PI(4)P at the Golgi. A mutant for PI4KIII β or PI4KII α should in principle, if the assumption that Golgi is the main source of PI(4,5)P₂ replenishment, prevents the regeneration of PI(4,5)P₂ at PM. This would add a genetic evidence for Golgi PI(4)P.

Response: We agree that this is an important issue should be clarified. To further confirm that the production of PI(4)P at the Golgi is the main source of PI(4,5)P₂ regeneration at the plasma membrane, we conducted PI4KII α knockdown to reduce PI(4)P generation in the Golgi of ORP4L KI T-cells. As shown in Figure 1g of the new version, depletion of PI4KII α prevented the recovery of PI(4,5)P₂ at the plasma membrane upon anti-CD3 stimulation, further supporting that PI(4)P at Golgi is the main source of PI(4,5)P₂ replenishment in the plasma membrane of ORP4L KI T-cells.

3- In the same way, given that it was shown before that PI4KIII β fuel OSBP by spatial proximity, we can expect that ORP4L acts in PI(4)P transfer not only by interacting with OSBP but maybe also through regulating PI4KIII β spatial proximity. Was PI4KIII β ever tested in co-IPs experiments as in Fig. 2?

Response: This indeed is an important question. As your suggestion, we conducted additional co-IP and immunofluorescence microscopy to investigate whether ORP4L also interacts with PI4KIII β or PI4KII α . PI4KIII β or PI4KII α could not be co-immunoprecipitated with ORP4L in the presence or absence of anti-CD3 stimulation in ORP4L KI T-cells (Supplementary Figure 1n of new version). In addition, both PI4KIII β and PI4KII α localized in the Golgi of ORP4L KI T-cells, and they did not translocate to the plasma membrane upon anti-CD3 stimulation (Supplementary Figure 1o of new version). Thus, we concluded that ORP4L did not regulate spatial PI4KIII β or PI4KII α proximity in ORP4L KI T-cells.

4- Interaction between OSBP and ORP4L was tested in co-IP. However, what is the in vivo evidence for direct interaction between OSBP and ORP4L? If in vivo experiment is lacking I suggest to be more cautious in the conclusion on OSBP and ORP4L interaction.

Response: Thank you for your this kind comment. To directly visualize the interaction of ORP4L

and OSBP *in vivo*, we employed the Bimolecular Fluorescence Complementation (BIFC) Assay (T.K.Kerppola et al., Nat.Protoc., 2006). Expression vectors encoding ORP4L/pVn-C1 and OSBP/pVc-C1 or ORP4L/pVn-C1 and OSBP/pVc-C1 were co-transfected into ORP4L KI T-cells. After 36 hr, heterodimer fluorescence was observed in the plasma membrane of cells transfected with ORP4L/pVn-C1 and OSBP/pVc-C1, moreover, this fluorescence signal was enhanced when the cells were stimulated with anti-CD3 (Figure 2f, g of new version), suggesting a specific interaction of ORP4L and OSBP *in vivo*.

5- I found very convincing that PI(4)P exchange against sterols occurs between Golgi and PM . However this conclusion only relies on the levels of PI(4)P at Golgi and PM in mutants. An *in vitro* lipid transfer assay would definitely strengthen this major point of the paper if this is technically and timely feasible in the frame of this study.

Response: It would for sure be helpful to conduct a lipid transfer assay by reconstituted membrane to strengthen our conclusions. But unfortunately, we cannot at the moment establish a system that reflects this unexpected non-vesicular and membrane contact site independent PI(4)P transfer mode. During our efforts to set up a reconstituted membrane assay, we found the following: 1) In this non-vesicular lipid transport machinery, although the gap between Golgi and PM is narrow, the organelles are not close enough to form MCSs; 2) We do not know whether some additional proteins are required for the formation of a narrow gap between Golgi and PM in this special cell type. At MCSs, tethering proteins such as VAPA are needed in artificial liposomes (Bruno Mesmin et. al., Cell, 2014), but we found VAPA is not involved in our transfer mode. In our artificial liposomes, ORP4L and OSBP could not make them tether together close enough to simulate the intracellular spatial positioning of the organelles. Thus, at this moment, we can only provide experimental evidence relying on *in vivo* data.

6- A previous study from the group shows that ORP4L might act to better present PI(4,5)P₂ for catalysis by PLC at the ER/Golgi interface. In the present work, activation of the AKT pathway is suggested to act downstream of ORP4L in T-cell leukemia. It is known that PLC controls AKT phosphorylation. So, at Golgi/PM interface in this cell type, an important question will be to determine whether the phosphorylation status of AKT is changed in ORP4L mutants, and would it depends then on PLC?

Response: PI(4,5)P₂ is a substrate for both PI3K and PLCβ3 catalysis, for downstream signaling activity. To further clarify the relationship of PLCβ3 and AKT activation, we further conducted ORP4L and PLCβ3 genetic modifications in ORP4L KI T-cells and the MT-4 cell line. We found ORP4L knockdown reduced AKT phosphorylation, but PLCβ3 knockdown did not change this status (Supplementary Figure 4d, e of new version). Moreover, in the rescue experiments, we found both wild-type ORP4L and a mutant ORP4L construct without the PLCβ3 binding site can abolish the reduction of AKT phosphorylation upon ORP4L knockdown in MT-4 cells (Supplementary Figure 4f of new version). Thus, we concluded the AKT activity is not dependent on PLCβ3 in this leukemia cell type.

Minor points

1- Oscillation or waves of OSBP/PI(4)P has been suggested to act at ER/Golgi interface, I found this

was a bit missing in the discussion of the present study for the Golgi/PM interface in T-cell.

Response: Thank you for the constructive criticism, we have stressed this in our discussion section.

2- Is it possible to add some kind of quantification for Fig. 2c, d, f, h?

Response : To understand the OSBP translocation more clearly, we quantified the OSBP fluorescence between plasma membrane and Golgi, and calculated their ratio. The results are shown in the corresponding figures of the new manuscript version.

Reviewer #2 (Remarks to the Author); expert in ATL:

Zhong et al. reported that expression of ORP4L is enhanced in adult T-cell leukemia (ATL) cells, and ORP4L associates with OSBP, resulting in translocation of OSBP from the Golgi apparatus and the plasma membrane (PM) to exchange PI(4)P and cholesterol. PI(4)P at the PM leads to PI3K/AKT hyperactivation, and malignant transformation of T cells. This study shows that abnormal lipid transport from Golgi to PM causes activation of PI3K/AKT, and NFkB, and eventually development of T-cell leukemia. Thus, this study reveals important roles of lipid metabolism and malignant transformation of T cells.

Major concerns

1. The same authors have recently reported that ORP4L interacts with PI3K δ to promote PI(3,4,5)P₃ generation and AKT activation in Blood. Authors should clarify the differences of these studies and discuss about discrepancies.

Response: Our recent study published in Blood reported that ORP4L is required for HTLV-1 oncogene Tax-induced T-leukemogenesis (Zhong et. al., Blood, 2022). We used ORP4L knock-out mice and demonstrated ORP4L deletion blocks Tax-induced T-cell leukemia. We studied how HTLV-1 induced ORP4L expression and found that HTLV-1 caused epigenetic loss of miR-31 resulting in the release of ORP4L expression. For molecular insight, ORP4L interacted with and activated PI3K δ , leading to PI(3,4,5)P₃ generation and AKT activation. In the Blood paper, we did not investigate the role of OSBP in ORP4L induced T-cell leukemogenesis, nor did we study how the homeostasis of PIPs in the plasma membrane is maintained and the substrate, especially PI(4)P, is supplied to support sustained robust PI3K/AKT signaling. In this study, we focused on how ORP4L regulates a dynamic equilibrium of PM PI(4)P and PI(4,5)P₂. By hiring T-cell specific ORP4L knock-in mice, we provided in vivo evidence that an unexpected non-vesicular lipid transport machinery established by ORP4L/OSBP protein complex is sufficient to drive malignant T-cell transformation and leukemogenesis. Thus, although some of the downstream effector AKT activation involved in these two studies, the focal points of the two studies are completely different. According to your suggestion, we have clarified these issues in the introduction section of new manuscript version.

2. This study showed that leukemic cells from ORP4L KI mice are CD4+CD25+, which resemble to ATL cells. However, CD4+CD25+ means that these cells are similar to activated CD4+ T cells or regulatory T cells. ATL cells are derived from dysregulated regulatory T cells. Authors should analyze expression of Foxp3, CCR4, and CD45RO of leukemic cells from ORP4L KI mice.

Response: According to your suggestion, we further identify the leukemia cells by analyzing the

Foxp3 and CCR4 expression in ORP4L KI T-cells. The percentage of CD25⁺Foxp3⁺ and CD25⁺CCR4⁺ T-cells were significantly increased in ORP4L KI T-cells (Figure 4d of new version). The Foxp3 and CCR4 expression were also upregulated in ORP4L KI T-cells (Figure 4e of new version). Because we were not able to obtain anti-mouse CD45RO antibody, we did not detect expression of this protein. In addition, the new blood smear from ORP4L KI mice showed cleaved 'flower cells', morphologically similar to those found in human ATL (Figure 4f, Figure 6g, i of new version), further supporting the presence of T-leukemia cells in ORP4L KI mice.

3. The quality of blood smear is too poor (Figures 4 and 5). It is impossible to judge these cells as leukemic cells.

Response: Thank you for your kind criticism. We have stained and replaced these poor blood smear images. We now present the leukemic cells in ORP4L KI mice with cleaved nuclei morphology, morphologically identical to the 'flower cells' in human ATL (Figure 4f, Figure 6g, i of new version).

4. Different from leukemia observed in ORP4L KI mice, the feature of skin lesion in ATL is the presence of leukemic cells in the epidermis. Authors should clarify whether leukemic cells infiltrate into the epidermis.

Response: Thank you for this suggestion. In this new manuscript version, we conducted new sectioning and staining of skin to rule out the present of skin disease in ORP4L KI mice. The skin from ORP4L KI mice revealed marked hyperkeratosis and acanthosis. Moreover, based on immunohistochemical staining, we found CD3⁺ T-cells infiltrated into the epidermis (Figure 4i, Supplementary Figure 3e of new version).

5. Authors found that "transcriptional misregulation in cancer || pathway" was affected in leukemic cells in ORP4L KI mice. However, since "transcriptional misregulation in cancer pathway" is too vague, authors should clarify whether dysregulated pathways in ATL (reference 38) are affected in this mouse model.

Response: Thank you for this excellent comment. According to the RNA sequencing data of the reference (Kataoka K, et al., Nat Genet., 2015), they found Antigen receptor signaling/NF-κB pathway as the major dysregulated pathway in human ATL. However, we did not find the same pathway by KEGG pathway analysis in T-cells of ORP4L KI mice. In our data, the "transcriptional misregulation in cancer" is mainly affected; We have now showed and labeled the misregulated genes of this pathway in Figure 5e of the new version. Some of these genes such as Cebpe (J L Wiemels et al., Leukemia, 2016), Erg (Shinobu Tsuzuki, Blood, 2011), Igf1 (Vincenzo Giambra et. al., Cell stem cell, 2018), Mmp9 (Javier Redondo-Muñoz et. al., Cancer cell, 2010), Mpo (Mohsen Hosseini et. al., Cancer research, 2019), Pax5 (A E Teo et. al, Leukemia, 2016), Prom1 (Laura Godfrey et. al., Leukemia, 2021) are well known to be involved in leukemia formation. We also wished to compare whether there is overlap in the dysregulated genes between ORP4L KI mice and human ATL as reported in this reference, but we could not download the raw data from database because we don't have permission to access the data.

6. In line 247-249, "Human ATL is associated with elevated PI3K/AKT activity, resulting in oncogenic properties through NF-κB dependent p53 inhibition (reference 39 and 40)". These findings are not confirmed by the following studies. Therefore, it is controversial among HTLV-1 researchers.

Response: This is an excellent suggestion. Actually, Akt and NF- κ B are frequently coordinately activated and can be affected by upstream molecules such as PI(3)K (Andrea Oeckinghaus et al. Nature Immunology, 2011), and the oncoprotein Akt controls NF- κ B activity in cancer cells (Han C. Dan et al. Genes & Development, 2008). According to your comment, to stress the link of ORP4L mediated AKT/NF- κ B/p53 signaling, we employed the AKT inhibitor LY294002 and the NF- κ B inhibitor TPCA-1 to treat ORP4L KI T-cells and the MT-4 cell line. AKT inhibition resulted in suppression of p65 and p53 phosphorylation; Meanwhile, also NF- κ B inhibition resulted in suppression of p53 phosphorylation (Supplementary Figure 4a of new version). Together with the previous observation that the role of AKT in NF- κ B/p53 signaling of HTLV-1 transformed cells (Soo-Jin Jeong et al. Oncogene, 2005), we speculated that in ORP4L KI T-cells, the enforced ORP4L expression leads to plasma membrane phospholipid dysregulation resulting in AKT and NF- κ B activation and p53 inhibition.

7. It is not generally accepted that “Transition from IL-2-dependent to independent growth is a key step in the transformation T-cells (line 268-269)”. Most of primary ATL cells do not proliferate by IL-2. This transition is observed only in in vitro culture system of ATL cells.

Response: HTLV-1 infected T-cells exhibit an initial phase of IL-2-dependent growth; over time, the cells become IL-2 independent (Migone T S, et al., Science, 1995). We have rewritten this sentence in our new manuscript. In addition, to confirm the T-cells transformation upon enforced ORP4L expression in vitro, we also detected Foxp3 and CCR4 expression. The transformation was further demonstrated by enhanced expression of these two proteins in ORP4L expressing T-cells (Figure 6b of new version).

Minor concerns

1. In line 96-99, “promoting” should be “promotes”.

Response: We have corrected this spelling mistake.

REVIEWERS' COMMENTS

Reviewer #1 (Remarks to the Author):

The manuscript from Zhong and collaborators describes a solid set of experiments suggesting the existence of a PI(4)P/sterol-exchange mechanism between the Golgi apparatus and the plasma membrane as a main source for PI(4,5)P₂ during AKT oncogenic signaling.

In my previous report I found the results very convincing and supportive of the conclusions drawn, I however raised some major and minor points to be solved. Satisfactorily, all my points were addressed by convincing discussions and most importantly new experiments. The authors now added new genetic and biochemical evidences to demonstrate that 1) the production of PI(4)P at the Golgi is the main source of PI(4,5)P₂ regeneration at the plasma membrane (using PI4KII α knockdown), 2) neither sphingomyelin nor the phospholipase PLC β 3 is involved in this process, 3) ORP4L did not regulate spatial PI4KIII β or PI4KII α proximity in ORP4L KI T-cells, 4) ORP4L and OSBP interact in vivo. Altogether, the results are now packed tightly and reinforce the conclusions drawn in this study. Thus, I do not have further comments to add on this very nice piece of work.

Reviewer #2 (Remarks to the Author):

In this revised manuscript, authors responded to the comments from Reviewer. The finding that Foxp3+ T cells in ORP4L KI mice are increased indicates the similarity between ORP4L KI T-cells and ATL cells. It suggests that activation of PI3K/AKT pathway contributes not only to leukemogenesis, but also to immunophenotype of ATL cells.

Reviewer comments:

Reviewer #1 (Remarks to the Author):

The manuscript from Zhong and collaborators describes a solid set of experiments suggesting the existence of a PI(4)P/sterol-exchange mechanism between the Golgi apparatus and the plasma membrane as a main source for PI(4,5)P₂ during AKT oncogenic signaling.

In my previous report I found the results very convincing and supportive of the conclusions drawn, I however raised some major and minor points to be solved. Satisfactorily, all my points were addressed by convincing discussions and most importantly new experiments. The authors now added new genetic and biochemical evidences to demonstrate that 1) the production of PI(4)P at the Golgi is the main source of PI(4,5)P₂ regeneration at the plasma membrane (using PI4KII α knockdown), 2) neither sphingomyelin nor the phospholipase PLC β 3 is involved in this process, 3) ORP4L did not regulate spatial PI4KIII β or PI4KII α proximity in ORP4L KI T-cells, 4) ORP4L and OSBP interact in vivo. Altogether, the results are now packed tightly and reinforce the conclusions drawn in this study. Thus, I do not have further comments to add on this very nice piece of work.

Response: Thanks very much for your kindly and excellent comments that helped us readdress issues of major importance and to significantly improve the manuscript.

Reviewer #2 (Remarks to the Author):

In this revised manuscript, authors responded to the comments from Reviewer. The finding that Foxp3⁺ T cells in ORP4L KI mice are increased indicates the similarity between ORP4L KI T-cells and ATL cells. It suggests that activation of PI3K/AKT pathway contributes not only to leukemogenesis, but also to immunophenotype of ATL cells.

Response: Thanks very much for your kindly and excellent comments that helped us readdress issues of major importance and to significantly improve the manuscript.